# Heterogeneous Catalysts in N-Heterocycles and Aromatics as Liquid Organic Hydrogen Carriers (LOHCs): History, Present Status and Future

**DOI:** 10.3390/ma16103735

**Published:** 2023-05-15

**Authors:** Jinxu Zhang, Fusheng Yang, Bin Wang, Dong Li, Min Wei, Tao Fang, Zaoxiao Zhang

**Affiliations:** 1School of Chemical Engineering and Technology, Xi’an Jiaotong University, Xi’an 710049, China; 2Shaanxi Hydrotransformer Energy Technologies Co., Ltd., Xi’an 712000, China; 3SPIC Guangzhou Branch, Guangzhou 511458, China

**Keywords:** energy transformation, hydrogen storage technology, liquid organic hydrogen carriers, optimization strategies, future development

## Abstract

The continuous decline of traditional fossil energy has cast the shadow of an energy crisis on human society. Hydrogen generated from renewable energy sources is considered as a promising energy carrier, which can effectively promote the energy transformation of traditional high-carbon fossil energy to low-carbon clean energy. Hydrogen storage technology plays a key role in realizing the application of hydrogen energy and liquid organic hydrogen carrier technology, with many advantages such as storing hydrogen efficiently and reversibly. High-performance and low-cost catalysts are the key to the large-scale application of liquid organic hydrogen carrier technology. In the past few decades, the catalyst field of organic liquid hydrogen carriers has continued to develop and has achieved some breakthroughs. In this review, we summarized recent significant progress in this field and discussed the optimization strategies of catalyst performance, including the properties of support and active metals, metal–support interaction and the combination and proportion of multi-metals. Moreover, the catalytic mechanism and future development direction were also discussed.

## 1. Introduction 

With the continuous reduction of fossil fuels and the rising public awareness of environmental protection, it is urgent to introduce clean and renewable energy to replace conventional fossil energy, so as to realize the transformation of the social energy structure and living style [1,2,3,4,5]. The production methods of hydrogen can be roughly divided into the following three types: hydrogen production from fossil fuels, hydrogen production from water electrolysis and hydrogen production from biomass [6,7]. Hydrogen production from fossil fuels is currently the mainstream method of hydrogen production, which is typically based on the pyrolysis, gasification and catalytic reforming processes of fossil fuels [8]. However, this method has disadvantages, such as a low energy conversion rate and serious environmental pollution, making it difficult to meet the goal of clean and efficient hydrogen production [6]. Hydrogen production from water electrolysis has good hydrogen production efficiency (generally 75.85%) and purity (above 99%) [6]. In addition, “green electricity” generated from renewable energy sources such as wind energy and solar energy can be used in this electrolysis process to achieve green hydrogen production [9,10,11]. Biomass hydrogen production uses biological processes such as fermentation and biophotolysis to produce hydrogen, with an efficiency of approximately 6–86%, and its main source of energy is solar energy [12]. Green hydrogen produced by electrolysis of water (using green electricity) and biomass is a clean and environmentally friendly energy carrier with great potential [13,14,15]. In addition, hydrogen energy has many advantages, such as a high energy density, high conversion efficiency, extensive sources and various forms of utilization, which also makes it an important energy alternative to achieve low-carbon development and energy transformation [16,17,18]. At present, hydrogen energy is widely used in many fields, such as distributed power generation, fuel cell vehicles, energy supply and so on [19,20,21].

The hydrogen energy industry chain mainly includes four aspects: hydrogen production, storage, transportation and utilization [22,23,24]. Providing inexpensive, efficient and safe hydrogen energy storage and transportation is one of the key issues restricting the development of hydrogen energy [8,25,26]. At present, the mature and widely used hydrogen storage technologies are mainly compressed gaseous hydrogen technology and low-temperature liquefied hydrogen technology. However, both need to consume a significant amount of energy (e.g., 31–43 kJ/mol H_2_ for compressing hydrogen to 800 bar, 28.4 kJ/mol H_2_ for liquefied hydrogen) [27] and cause safety concerns [28,29,30]. Therefore, researchers are committed to developing new, more efficient and economical hydrogen storage technology, including metal hydride hydrogen storage [31,32], porous materials hydrogen storage [33,34], complex hydride hydrogen storage [35,36] and liquid organic hydrogen carrier hydrogen storage [37,38,39]. Liquid organic hydrogen carrier technology uses hydrogen-deficient carriers of liquid organics to store hydrogen through a catalytic hydrogenation reaction. When hydrogen is needed, hydrogen is released from hydrogenated carriers through a catalytic dehydrogenation reaction. During the reversible storage of hydrogen, liquid organic hydrogen carriers (LOHCs) themselves hardly consume [10,40]. Liquid organic hydrogen carrier technology has the advantages of a high hydrogen storage density, good reversibility, satisfactory safety and good compatibility with existing energy transportation logistics infrastructure, and is widely considered as a promising hydrogen storage technology [41,42,43,44].

The idea of using liquid organic compounds for the cyclic storage of hydrogen through a chemical reaction was first proposed by Sultan and Shaw [45] in 1975. At the beginning, research on liquid organic hydrogen carriers mainly focused on finding suitable compounds from liquid aromatic hydrocarbons as hydrogen storage carriers, such as benzene, toluene and naphthalene. However, the high dehydrogenation reaction temperature and slow reaction rate of these aromatic compounds limit their large-scale application [46,47]. In the follow-up research, scholars have been committed to developing novel liquid organic hydrogen carriers that might realize reversible hydrogen storage under relatively mild temperature conditions. From 2004 to 2008, Pez and Cheng reported a series of heteroaromatic compounds with low melting points and dehydrogenation temperatures (<200 °C) in their published patents [48,49,50,51,52]. Among these heteroaromatic compounds, N-ethylcarbazole (NEC), with a high hydrogen storage density of 5.8 wt%, has attracted extensive attention and is considered to be a promising carrier material for liquid organic hydrogen [46,53,54]. Pez et al. effectively reduced the dehydrogenation reaction temperature and energy requirements of liquid organic hydrogen carriers by introducing heteroatoms (N, S…) into traditional aromatic compounds, opening up a very promising way for the development of organic liquid hydrogen carriers. After that, heteroaromatic compounds became a hot field of research for new organic liquid hydrogen carriers, and more and more heterocyclic compounds with a high hydrogen storage density have been developed, such as N-propylcarbazole (NPC) [55,56], 1-methylindole (1-MID) [57], 2-methylindole (2-MID) [58], 1,2-dimethylindole (1,2-DMID) [10], N-ethylindole (NEID) [59,60], 7-ethylindole(7-EID) [61], 2-(N-methylbenzyl)-pyridine (MBP) [62] and acridine (ACD) [47], etc. In addition, dibenzyltoluene (DBT), with a high hydrogen storage capacity and suitable melting and boiling points, was first used as the liquid organic hydrogen carrier by Brückner et al. [63] in 2014 and has been studied since then [38,64,65]. Table 1 lists the physical property parameters and mass hydrogen storage density data of typical liquid organic hydrogen carriers (hydrogen-deficient carriers).

When the targets of a high hydrogen storage density (>5.5 wt%) and low dehydrogenation temperature (<200 °C) for LOHCs are achieved simultaneously, another key issue for the large-scale commercial application of liquid organic hydrogen carrier technology is to realize the reversible and efficient hydrogenation reaction and dehydrogenation reaction process of LOHCs. In the processes of hydrogen storage and release of LOHCs, the hydrogenation reaction is exothermic and high conversion rates can only be achieved at lower temperature levels [66]. The reaction pressure is usually between 5–7 MPa. A reaction pressure that is too low is not conducive to the reaction rate, and a reaction pressure that is too high will promote the occurrence of side reactions [67]; a dehydrogenation reaction is an endothermic reaction that is usually carried out under atmospheric pressure. The temperature of the dehydrogenation reaction needs to be controlled within an appropriate range. A dehydrogenation temperature that is too low will lead to a lower reaction rate. Although a high dehydrogenation temperature is thermodynamically beneficial, it will consume a lot of energy, which is not conducive to the commercial application of liquid organic hydrogen carriers. Under the conditions of an appropriate reaction temperature and pressure, the catalyst with high catalytic activity, selectivity and good stability becomes the key to realizing the efficient hydrogenation reaction of the liquid organic hydrogen carrier, and realizing the large-scale commercial application of liquid organic hydrogen carrier technology is also one of the core issues to consider [68,69]. In this paper, the research on hydrogenation and dehydrogenation catalysts of LOHCs will be reviewed, and future research directions will be considered, hoping to help readers quickly and clearly understand the research progress in this field.

## 2. Single-Function Catalysts of LOHCs

Over the past few decades, catalysts for the hydrogenation and dehydrogenation of liquid organic hydrogen carriers have been extensively studied. Up to the present, most relevant catalysts have been used with a single function, which means they can work on either hydrogenation or dehydrogenation, but not both [67]. After numerous experiments, researchers found that the catalysts with noble metals, such as Ru [70,71,72], Pd [73,74,75], Pt [76,77,78] and Rh [79,80], showed excellent performance in the hydrogenation and dehydrogenation of LOHCs. In addition, researchers also found that selecting an appropriate catalyst carrier to load active metal components can further improve the catalytic performance of the catalyst [81,82,83]. In the following sections, research developments in relation to catalysts of LOHCs will be elaborated from the aspects of active metal components and carriers. 

### 2.1. Hydrogenation Catalysts of LOHCs

#### 2.1.1. Monometallic Hydrogenation Catalysts

The liquid organic hydrogen carrier materials developed early, e.g., benzene, toluene and naphthalene, are basic and important raw materials in the petrochemical industry, whose products after hydrogenation are widely used and of high industrial value [84,85]. Up to now, the hydrogenation of benzene, toluene and naphthalene has been conducted with mature technology, and the corresponding catalysts have been extensively studied. Most of the active metals in their hydrogenation catalysts come from Group VIII (Table 2), such as Ni [86,87], Pt [88,89], Pd [90,91] and Ru [72,92], etc.

Up to now, researchers have developed some catalysts with exciting catalytic properties of benzene, toluene and naphthalene. For example, Su et al. [93] used surface carbon-coated SBA-15 (C-silica) as the catalyst carrier and prepared a 4.2 wt% Ru/C-silica catalyst via the thermal reduction method. Its TOF value in the process of benzene hydrogenation is as high as 37,700 h^−1^, which is almost the most active catalyst at present. The excellent catalytic activity of the Ru/C-silica catalyst may be due to the interface interaction between metal R and carbon support and the formation of Ru-C chemical links. In the process of thermal reduction, Ru nanoparticles are semi-embedded in the carbon matrix, making the surface contact between Ru and carbon more extensive, which may promote the transport of hydrogen spillover species, thus promoting the hydrogenation reaction. In addition, the chemical bond of saturated C atoms in the graphene layer after high temperature heat treatment is destroyed. Then, these unsaturated C atoms form Ru-C chemical bonds with Ru atoms (Figure 1), enhancing the stability of Ru particles and promoting the transfer of hydrogen spillover species and electrons between the active component Ru and the carrier surface. Song et al. [96] creatively prepared nickel nanocrystals with three-dimensional hierarchical flower-like structures (Figure 2), which could realize the complete hydrogenation of toluene within 30 min. The unique three-dimensional hierarchical flower structure makes nickel nanocrystals have a large surface area and abundant exposed planes, significantly improving the catalytic performance. In addition, nickel nanocrystals have good stability, with the catalytic activity retained above 95% after six catalytic experiments.

In recent years, the monometallic hydrogenation catalysts of LOHCs mostly focus on n-heterocyclic compound systems proposed by Pez and his colleagues, and NEC is the research focus among them [68,82,100,101,102]. The melting point of NEC at an atmospheric pressure is about 68 °C, so it is in a solid state at room temperature. Eblagon et al. [103] investigated the hydrogenation reaction of NEC in a molten state for the first time, exploring its hydrogenation reaction on a variety of noble metals and nickel-based catalysts. Their experimental results show that the conversion rate and product selectivity of the hydrogenation reaction of molten state NEC depend on both the metal active components and the types of carriers. On the same activated carbon carrier, the catalytic activity of different metal components follows the order: Ru > Pd > Pt > Ni. Among a series of Ruthenium-based catalysts with different carriers, a 5 wt% Ru/Al_2_O_3_ catalyst showed the best catalytic activity. In order to further explore the reasons for the difference in the catalytic activity of NEC hydrogenation between different metal components, Eblagon et al. [106] subsequently studied the hydrogenation of NEC in a molten state on unsupported noble metal (Ru, Pd. Pt) and supported Ni catalysts, confirming the outstanding activity of Ru. They found that the difference in the catalytic reaction activity and product selectivity of different metals mainly depends on their electronic structure. The hydrogenation rate of NEC decreased with the increase in the d-band center of the active metal in the catalyst, hence the lowest d-band center position of metal Ru among candidates under discussion may account for the highest catalytic activity. Moreover, Eblagon et al. [79] also studied the hydrogenation of NEC catalyzed by Ru-based, Pd-based and Rh-based catalysts, in which three different catalyst supports, i.e., activated carbon (AC), rutile (TiO_2_) or alumina (Al_2_O_3_), were used. Their experimental results showed that in the liquid-phase hydrogenation of ethyl carbazole, no matter which carrier was used, Ru was the most active metal, followed by Rh and Pd. In addition, the authors also found that the structure and composition of the product of dodecahydro-N-ethylcarbazole (12H-NEC) were mainly affected by the type of catalyst carriers. The presence of hydrophilic catalyst carriers, such as Al_2_O_3_ and TiO_2_, can promote the formation of cis-isomers of 12H-NEC, which is more advantageous in the dehydrogenation process. These studies have comprehensively proved that metal Ru has high catalytic activity for the hydrogenation of NEC.

With the discovery of more N-heterocyclic compounds, Ru-based supported catalysts have become the most commonly used hydrogenation catalysts of LOHCs, which are often used to study the hydrogen storage performance, optimal reaction conditions and hydrogenation kinetics of new organic liquid hydrogen carriers [47,55,57,58,60]. Dong et al. [60] studied the hydrogenation reaction of N-ethylindole over the 5 wt% Ru/Al_2_O_3_ catalyst at 160–190 °C and 9 MPa. The experimental results showed that N-ethylindole can be completely hydrogenated within 80 min at 190 °C. Li et al. [58] discovered a new liquid organic hydrogen storage carrier—2-methylindole. Between the temperature of 120–170 °C and under the pressure of 7 MPa, the hydrogenation of 2-methylindole was catalyzed by a 5 wt% Ru/Al_2_O_3_ catalyst. The experimental results showed that at 160 °C, 2-methylindole was completely hydrogenated in 40 min. Yang et al. [55] studied the hydrogenation reaction of NPC in a molten state over a Ru/Al_2_O_3_ catalyst at 120–150 °C. The experimental results showed that at the optimum temperature of 120 °C, the molten NPC could be completely hydrogenated within 60 min. Importantly, there were two isomers in the hydrogenation products. At a lower temperature and/or with a shorter reaction time, the less stable stereoisomer that is conducive to the release of hydrogen during dehydrogenation accounts for the majority of the products. The authors believed that low-temperature hydrogenation not only improved the rate of reaction, but also facilitated the subsequent dehydrogenation process.

The catalyst carrier is one of the important factors that affect the performance of a supported catalyst. The catalyst carrier can modify the interaction between the reaction substrate and the metal active component, and the appropriate carrier can effectively enhance the catalyst activity [81,82,83,100]. In order to further improve the catalytic performance of hydrogenation catalysts, researchers had attempted to use new catalyst carriers instead of conventional Al_2_O_3_ to support metal ruthenium, and obtained exciting results. Qin et al. [100] used partially graphitized biocarbon (pg-BC) as the support and prepared a highly active RuNPs/pg-BC catalyst via the wet impregnation carbothermal reduction method, in which the theoretical load of metal Ru was 5 wt%. At 130 °C and a pressure of 6 MPa H_2_, the hydrogenation of NEC was carried out with a RuNPs/pg-BC catalyst. The conversion of NEC reached 100% within 70 min, and the selective yield of 12H-NEC reached 99.41%. In addition, after nine consecutive catalytic reactions, the RuNPs/pg-BC catalyst could still maintain a 100% NEC conversion rate and 92.54% 12H-NEC selectivity, exhibiting good stability. The characterization results showed that carbon atoms in the biochar were consumed during the thermal reduction process, thus producing many cavities on its surface. The high stability of the catalyst was caused by the Ru nanoparticles embedded in these cavities (Figure 3). Moreover, the partial graphitization of physical carbon enhances the conductivity and electron transfer ability of the support, which is conducive to the improvement of catalytic activity [107,108]. In order to achieve better catalytic hydrogenation performance, Wu et al. [102] innovatively used rare-earth hydride as the hydrogenation catalyst carrier, and successfully developed a Ru/YH_3_-supported catalyst for the hydrogenation of NEC. The complete hydrogenation of NEC over the Ru/YH_3_ catalyst can be realized at 90 °C and a 1 MPa hydrogen pressure, which is the lowest hydrogenation pressure of ethyl carbazole reported at present. A 1.3 wt% Ru/YH_3_ catalyst not only reduces the use of noble metals, but also maintains high catalytic activity. Its TON value (the mass of substances completely converted per mole of active metal per hour, which is calculated according to the hydrogenation curve) was as high as 318 h^−1^, much higher than many catalysts under the same reaction conditions. Further research indicated that this outstanding activity was due to the reversible hydrogen absorption and desorption of YH_3_ that provides a new hydrogen transfer path: the atomic hydrogen in YH_3_ is transferred to the activated NEC molecule on the catalyst, and then the hydrogen vacancy left in YH_3_ is supplemented by H_2_. To verify the assumption experimentally, Ru/YH_3_ was used for the hydrogenation reaction in the absence of a hydrogen supply, and the reaction temperature was lower than the initial temperature of YH_3_ decomposition. The nuclear magnetic resonance spectra of the products indicate the presence of partial hydrogenation products, confirming the existence of the proposed hydrogen transfer pathway. In addition, the Ru/YH_3_ catalyst showed significant stereoselectivity for the all-cis hydrogenation products, and the selectivity of all-cis 12H-NEC in the reaction products was up to 91 mol%, which was significantly higher than the 70 mol% of the ruthenium black catalyst [79]. The steric hindrance of the all-cis products was small, and dehydrogenation reactions could easily occur. Therefore, the mixed products of the all-cis products with a high proportion were conducive to the subsequent dehydrogenation process. Moreover, the Ru/YH_3_ catalyst could also efficiently catalyze the hydrogenation of 2-methylindole and other rare earth hydride-supported catalysts, for e.g., Ru/LaH_3_ and Ru/GdH_3_ also showed good activity for the hydrogenation of NEC, which confirmed that a rare-earth hydride-supported catalyst was a feasible strategy to develop an efficient n-heterocyclic hydrogenation catalyst.

Although Ru-based catalysts have good catalytic hydrogenation performance, the high cost limits their wide application. In order to improve the economy of the catalyst, many researchers are dedicated to developing non-noble metal based catalysts, and Ni-based catalysts have become the most studied candidate for the hydrogenation of LOHCs [97,98,101]. The catalytic activity of a traditional Ni/Al_2_O_3_ catalyst for the hydrogenation reaction of LOHCs is low, which is due to the formation of a low-reduction NiAl_2_O_4_ spinel structure, thus significantly reducing the reducibility of Ni [109,110]. In order to improve the catalytic performance of nickel-based catalysts, Wu et al. [97] successfully prepared a Ni/Al_2_O_3_-YH_3_ catalyst by simply mixing a rare-earth hydride YH_3_ and Ni/Al_2_O_3_ catalyst in a mass ratio of 1:4. Their experimental results showed that the catalytic performance of the Ni/Al_2_O_3_-YH_3_ catalyst for the hydrogenation of NEC was comparable to that of the noble metal Ru/Al_2_O_3_ catalyst, achieving great improvement compared to Ni/Al_2_O_3_. Similar to the Ru/YH_3_ catalyst, the addition of YH_3_ provides a new hydrogen transfer pathway (see isotope labeling results in Figure 4), significantly improving the catalytic performance. Surprisingly, they found that the metal hydrides added can also promote the dehydrogenation process of LOHCs, which provides a potential path for the development of bifunctional catalysts for both the hydrogenation and dehydrogenation of liquid organic hydrogen carriers. Ding et al. [98] synthesized a series of Ni-based catalysts with Al_2_O_3_, Al_2_O_3_-SiO_2_ and SiO_2_ as carriers via a co-precipitation method, using the prepared catalyst for the hydrogenation of LOHCs with great application potential, such as benzene, NPC, NEC and DBT. The experimental results showed that the Ni_70_/AlSiO-1/1 catalyst with a molar ratio for Al and Si of 1:1 in the support had the best catalytic effect. The catalyst exhibited strong adaptability to different LOHCs; high catalytic activity in the hydrogenation reaction NPC, NEC and DBT and could realize the complete hydrogenation process of each LOHC within 1.5 h. In addition, Ni70/AlSiO has good stability. In the fifth hydrogenation cycle, NPC reached a hydrogen storage capacity of 5.43 wt% in 1.5 h. The characterization results show that the SiO_2_ component in the AlSiO composite support is conducive to the reduction of the active metal component Ni, while the presence of Al_2_O_3_ inhibits the aggregation of Ni and facilitates the dispersion of Ni. The Ni_70_/AlSiO-1/1 catalyst involves no use of precious metals and high-priced carriers, thus greatly reducing the cost. There is no doubt that the Ni_70_/AlSiO-1/1 catalyst with low cost, high catalytic activity and good adaptability to different LOHCs is highly suitable for large-scale commercial applications of LOHCs, proving the potential of the non-noble metal-based catalyst to replace the noble metal Ru-based catalyst. Rare-earth elements are plentiful in the earth’s crust, and China has abundant rare earth resources [111,112]. Therefore, rare-earth hydride catalysts with excellent catalytic hydrogenation performance have broad prospects in the field of organic liquid hydrogen storage, especially in the field of fixed hydrogen storage. However, compared to a conventional Al_2_O_3_ carrier, the cost of rare-earth hydride catalysts is still relatively high. Therefore, the development of low-cost catalysts with excellent catalytic performance can aim to use other affordable metal hydrogen storage materials (such as Mg_2_NiH_4_ and LaNi_5_) to replace rare-earth hydrides in the future.

Up to now, the development of high-performance and low-cost hydrogenation catalysts for aromatic and N-heterocyclic compounds as LOHCs mainly involves the use of new catalyst carriers (such as YH_3_ and pg-BC) or the addition of accelerators (such as YH_3_ and SiO_2_) to improve conventional carriers and the development of non-noble metal-based catalysts to replace noble metal-based catalysts. Moreover, the combination of the above two pathways is expected to develop more promising hydrogenation catalysts in the future.

#### 2.1.2. Bimetallic Hydrogenation Catalysts

Hydrogenation catalysts with high catalytic performance and of low cost have always been the research goal of scholars in the field of LOHCs, for which doping a second metal component into single metal catalysts, i.e., a bimetallic catalyst, is a promising strategy. A large number of bimetallic hydrogenation catalysts have been successfully developed and reported, mostly benzene, toluene and naphthalene. When N-heterocyclic compounds such as NEC with a high hydrogen storage density and mild dehydrogenation conditions were made available, the focus of research on bimetallic hydrogenation catalysts of LOHCs gradually turned towards the N-heterocyclic compound system. Table 3 summarizes the hydrogenation performance of the typical bimetallic catalysts of LOHCs.

Among the bimetallic catalysts, it is significant to select a bimetallic combination with an excellent catalytic performance. Yang et al. [91] synthesized a series of supported bimetallic Pd-M/SiO_2_ (M = Pt, Ni, Cu, Co) catalysts with different metal combinations via the simultaneous strong electrostatic adsorption (co-SEA) method; the catalysts were used in the hydrogenation of BEN and TOL to investigate their catalytic activity. The experimental results showed that the catalytic activity of the co-SEA Pd-Ni/SiO_2_ catalyst was almost twice that of the catalyst synthesized via the conventional dry impregnation method. The Pd 3d spectrum and H_2_-TPR results showed that the Pd component tended to provide electrons to the SiO_2_ carrier. In addition, Pd had low electronegativity, and the second component metals also tended to obtain electrons from Pd. Based on the above factors, the SEA Pd-M/SiO_2_ catalysts contained a large amount of Pd δ+ sites. The large number of electron-deficient Pd δ+ sites on the surface of SEA Pd-M/SiO_2_ that easily adsorb electron-rich benzene and toluene was the main reason for this beneficial result. In addition, the order of activity of the bimetallic catalysts under discussion is: Pd-Co/SiO_2_ < Pd-Cu/SiO_2_ < Pd-Pt/SiO_2_ < Pd-Ni/SiO_2_. The Pd-Ni/SiO_2_ catalyst with the highest degree of electron deficiency of the Pd component showed the highest catalytic activity for the hydrogenation of benzene and toluene.

With the continuous in-depth study of the hydrogenation catalysts of LOHCs, metal ruthenium has been proved by many research groups to be the most effective noble metal hydrogenation component [79,103,106]. Therefore, using other non-noble metals to replace part of ruthenium is a very promising method to achieve low costs and develop high-catalytic activity bimetallic catalysts. On the basis of a RuNPs/pg-BC catalyst supported by partially graphitized biochar, Qin et al. [118] introduced nickel metal as the second metal into the RuNPs/pg-BC catalysts and successfully prepared a series of Ni-Ru alloy nano-catalysts (Ni-RuNPs/pg-BC) with different Ni/Ru mass ratios. The experimental results of the catalytic hydrogenation of ethyl carbazole showed that the catalytic performance of the Ni_0.5_Ru_4.5_/pg-BC catalyst was the best. Interestingly, the catalytic activity of the Ni_0.5_Ru_4.5_/pg-BC catalyst was even higher than that of the Ru/pg-BC catalyst, while the amount of noble metal Ru was reduced by 10%. This result could be attributed to the synergism of Ni and Ru, making H_2_ easier to be activated and generating hydrogen spillover, promoting the adsorbed NEC molecules to be continuously hydrogenated. Therefore, the Ni_0.5_Ru_4.5_/pg-BC catalyst realized a two-fold optimization of both the catalytic activity and preparation cost. Wang et al. [120] synthesized a series of Ru-Ni alloy nano-catalysts (Ru_x_Ni_1−x_/SBA15) with different Ru/Ni ratios supported by SBA-15 via the electrostatic adsorption method, and they were tested for the catalytic hydrogenation of NEC. Their experimental results showed that the catalytic activity of Ru_x_Ni_1−x_/SBA15 formed by adding between 10–30% Ni was better than that of the Ru/SBA15 catalyst at 100 °C and 5 MPa H_2_ pressure. Among them, the Ru_0.7_Ni_0.3_/SBA15 catalyst exhibited the best catalytic activity, as is shown in Figure 5. At a temperature of 60 °C, the catalytic activity of NEC on Ru_0.7_Ni_0.3_/SBA15 was still better than that of the commercial Ru/Al_2_O_3_ catalyst at 90 °C. Moreover, the hydrogen storage rate of NEC in three consecutive catalytic hydrogenation reactions was higher than 99% for all reactions, with the catalysis of Ru_0.7_Ni_0.3_/SBA15. The characterization results of the Ru_0.7_Ni_0.3_/SBA15 catalyst showed that the good dispersion of the RuNi alloy caused via the electrostatic adsorption process and the hydrogen spillover effect from the RuNi alloy components improved the catalytic activity. On the other hand, the improvement of catalyst stability was due to the complexation of the RuNi alloy NPs with surface hydroxyl groups.

For bimetallic catalysts, the content of each active metal component affects its catalytic performance, thus the appropriate proportion of bimetallic composition is crucial for the development of highly active catalysts. Zhu et al. [113] prepared a series of Ru_x_Ni_y_/C catalysts with different Ru/Ni atomic ratios via the hydrazine hydrate reduction method and electric displacement method, which were used in benzene hydrogenation to evaluate the catalytic performance. The experimental results showed that the catalytic performance of different Ru/Ni atomic ratios was as follows: Ru_0.08_Ni_0.92_/C < Ru_0.16_Ni_0.84_/C < Ru_0.83_Ni_0.17_/C < Ru_0.34_Ni_0.66_/C < Ru_0.56_Ni_0.44_/C. We can note that the catalytic activity first increases and then decreases with the increase in the Ru/Ni atom ratio, reaching the optimum at 0.56/0.44. The characterization results showed that the high catalytic activity of the Ru_0.56_Ni_0.44_/C catalyst was due to the high synergistic effect of Ru, Ni and NiO sites produced by the nanostructure of Ru-on-Ni/NiO nanoparticles. Hydrogen molecules and benzene are adsorbed and activated on Ru sites and NiO sites, respectively, while Ni sites promotes the transfer of activated H from Ru sites to NiO sites. The low catalytic activity of the Ru_0.83_Ni_0.17_/C catalyst was caused by some factors harmful to the activation and dissociation of hydrogen, including the Ru component, whose main chemical state was ruthenium oxide (RuO_2_); in addition, RuO_2_ coated on the surface of NiO and large Ru-Ni BNPs formed. The reason for the low performance of the catalyst with a low Ru/Ni atomic ratio was that the synergistic effect of the Ru, Ni and NiO sites was not fully developed. Li et al. [121] prepared a series of 5 wt% Ru_x_Ni_5−x_/Al_2_O_3_ (x = 0, 1, 2.5, 4, 5) bimetallic catalysts via the liquid-phase reduction method and used them to catalyze the hydrogenation of NPC. The experimental results showed that the catalytic activity of RuNi/Al_2_O_3_ bimetallic catalysts with different proportions is as follows: Ru_2.5_Ni_2.5_/Al_2_O_3_ > Ru_4_Ni_1_/Al_2_O_3_ > Ru_5_/Al_2_O_3_ > Ru_1_Ni_4_/Al_2_O_3_ > Ni_5_/Al_2_O_3_. The 5 wt% Ru_2.5_Ni_2.5_/Al_2_O_3_ catalyst with the highest metal dispersion and the lowest average particle size had the best catalytic performance, achieving higher catalytic activity than the Ru_5_/Al_2_O_3_ catalyst with a 50 % reduction in Ru metal usage.

Similar to monometallic catalysts, the catalyst carriers are also significant for bimetallic catalysts. The strong interaction between the active metal components on the catalyst and the carrier can effectively improve the catalytic activity and stability. This is well proven by the research of Yu et al. [105]. For the hydrogenation system of NEC, they studied the influence of three different types of TiO_2_ carriers—rutile, anatase and commercial P25 (rutile/anatase—1/4 mixture)—on the catalytic performance of Ru monometallic and Ru-Ni bimetallic catalysts. Their results showed that the addition of Ni promoted the H spillover effect and effectively improved the performance of the Ru/anatase catalyst. Unfortunately, when rutile was used as the catalyst carrier, Ru and Ni would form serious metal aggregation, thus significantly reducing the selectivity of the Ru-Ni/rutile catalyst for complete hydrogenation products. For the Ru-Ni bimetallic component, the catalytic performance of the catalyst supported by the mixture of rutile/anatase—P25 was significantly better than that of the catalyst supported by rutile and anatase, as is shown in Figure 6, proving a certain promotion between the two different types of TiO_2_ crystals. 

Moreover, the multiple sites of the carrier and active metal components could cooperate to improve the catalytic performance. Zhu et al. [116] prepared a Ru/Ni/Ni (OH)_2_/C catalyst using Ni/Ni (OH)_2_ nanoparticles (NPs) as anchor carriers via a hydrazine hydrate reduction and electric replacement reaction, as is shown in Figure 7. The results of the catalytic hydrogenation of naphthalene showed good catalytic activity of the Ru/Ni (OH)_2_/C catalyst, which was significantly superior to the Ru/C, Ni/Ni (OH)_2_/C and Ru-Ni alloy/C catalysts. This remarkable activity was attributed to the interface synergy of the Ru, Ni and Ni (OH)_2_ sites. Hydrogen is first adsorbed and activated by Ru, and then the activated hydrogen ion is transferred by Ni to the activated naphthalene at Ni(OH)_2_ sites to generate decalin. In the follow-up study, Deng et al. [117] prepared Ru/Ni/NiO/C and Ru/Co/Co_3_O_4_/C catalysts with transition metal/transition metal oxide nanoparticles (TM/TMO NPs) as anchor carriers using the same method. These catalysts showed high catalytic activity and 100% decalin selectivity in a naphthalene hydrogenation reaction. Actually, the special nanostructures (Ru nanoclusters or single atoms supported on Ni/NiO (Co/Co_3_O_4_) NPs) in these catalysts promoted the synergistic effect among the multiple catalytic sites, thus improving the catalytic performance. This synergistic effect of the multiple active sites is shown in Figure 8, where naphthalene with negative charge is preferentially adsorbed electrophilically on TMO sites (NiO or Co_3_O_4_) with positive charge and activated, while H_2_ is activated at the Ru sites to generate activated H*. H* is transferred to the surface of NiO or Co_3_O_4_ through the hydrogen spillover effect of TM (Ni or Co) and then reacts with activated naphthalene to generate decalin. The research of Deng et al. has proven that the design of multiple active sites capable of synergistic action in the catalyst is a potential way to develop efficient hydrogenation catalysts.

#### 2.1.3. Other Hydrogenation Catalysts

Metal hydride, a hydrogen storage material with excellent hydrogenation/dehydrogenation kinetics [31,122,123], is also expected to become an efficient hydrogenation catalyst of LOHCs. Feng et al. [124] used a Mg-based metal hydride composed of Mg_2_NiH_4_, MgH_2_ and LaH_3_, which were prepared via reactive ball milling, as the catalyst for DBT hydrogenation for the first time. The experimental results showed that the catalytic activity of the Mg-based metal hydride after grinding for 500 min was the best, and the hydrogen absorption capacity of DBT reached 4.63 wt% after 4 h. Moreover, XRD results showed that the Mg-based metal hydride was composed of MgH_2_, Mg_2_NiH_4_ and LaH_3_. SAED results showed that Mg_2_NiH_4_ and LaH_3_ nanoparticles were uniformly dispersed in the MgH_2_ matrix. The results of the catalytic hydrogenation of MgH_2_, Mg_2_NiH_4_ and LaH_3_ showed that Mg_2_NiH_4_ was the main catalytic phase of the Mg-based metal hydride. Based on the above characterization results and relevant analysis, the catalytic hydrogenation mechanism proposed by Feng et al. is shown in Figure 9. In the Mg-based metal hydride, Mg_2_NiH_4_ on the MgH_2_ matrix provides the activation sites for DBT. Hydrogen molecules are absorbed by metal hydride and dissociated into two hydrogen atoms in the lattice which can move into DBT molecules and undergo hydrogenation reaction. Many catalytic active centers were observed to be formed on the surface of Mg_2_NiH_4_ nanoparticles, promoting the dissociation and diffusion of hydrogen. Moreover, the presence of LaH_3_ nanoparticles also provides additional active sites of hydrogen molecules, thus further improving the catalytic activity. Therefore, the Mg-based metal hydride had good catalytic performance. The research of Feng et al. provided a new method for the application of metal hydride and also contributed a potential strategy for the development of high-efficiency hydrogenation catalysts for LOHCs.

Inspired by the successful development of bimetallic catalysts with better catalytic performance than monometallic catalysts, scholars have turned their attention to the development of tri-metal catalysts. Zhu et al. [125] successfully prepared tri-metal nano-catalysts (Ru-NiCo/C unfired) and several bimetallic nano-catalysts through a hydrazine hydrate reduction and electric replacement reaction and evaluated the catalytic performance of these catalysts through a naphthalene hydrogenation reaction. At 100 °C and a hydrogen pressure of 4.48 MPa, the tri-metal catalyst (Ru-NiCo/C unfired) exhibited better catalytic performance than the bimetallic catalysts (Ru/Ni/Ni(OH)_2_/C and Ru/Co/Co (OH)_2_/C). Under the catalysis of the Ru-NiCo/C unfired catalyst, the hydrogenation of naphthalene could be completed within 30 min, and the selectivity of decalin could reach 100%. The characterization results showed that the better performance of the tri-metal catalyst came from the unique nanostructure of the Ru nanoclusters-on-NiCo-on-Ni (OH)_2_-Co(OH)_2_ in the catalyst, as well as the stronger synergistic effect among Ru, Ni and Co than that of two metals. Their research proved that on the basis of bimetallic catalysts, tri-metal catalysts could further improve catalytic performance. Therefore, building a synergistic effect among multiple metals is also a promising strategy for developing high-activity, low-cost catalysts.

### 2.2. Dehydrogenation Catalysts of LOHCs

#### 2.2.1. Monometallic Dehydrogenation Catalysts

Similar to hydrogenation reactions, noble metal-based catalysts are also widely used in the dehydrogenation reaction of liquid organic hydrogen carriers. For aromatic liquid organic hydrogen carriers such as benzene, toluene, naphthalene, etc. Pt-based catalysts usually show very good catalytic performance, because metal Pt can selectively catalyze the breaking of a C-H bond, while its ability to destroy C-C is weak [126]. Alumina (Al_2_O_3_) and activated carbon are common carriers of Pt-based catalysts [127,128,129]. In order to develop Pt-based catalysts with good dehydrogenation performance, in-depth research has been conducted [81,130,131,132]. Jiang et al. [127] studied the effect of platinum-based catalysts supported on alumina and carbon carriers on the dehydrogenation of decalin. The research results showed that the dehydrogenation activity of a Pt/AC catalyst for decalin was better than that of a Pt/Al_2_O_3_ catalyst, which was due to the hydrogen overflow effect of the Pt/AC catalyst and better dispersion of Pt. As is well known, different preparation methods will generally lead to differences in the physical properties of metal catalysts, such as the dispersion and particle size of metal components, which will greatly affect the catalytic performance [132,133]. Lee et al. [132] prepared carbon-supported platinum catalysts for the dehydrogenation of decalin using four different preparation methods: an impregnation method, precipitation method, ion exchange method and polyol method. The research results showed that the initial hydrogen evolution rate and total hydrogen evolution with Pt/C catalysts prepared via advanced methods (ion exchange method and polyol method) were higher than with those prepared via traditional methods (impregnation method and precipitation method), which could be attributed to the better platinum dispersion of the catalysts prepared via the former methods.

In the early stage, the platinum content of supported platinum catalysts was usually 5–10 wt% [134,135,136], bringing about rather high costs. To address this issue, on the one hand, scholars tried to replace the conventional carrier and strengthen the interaction between the metal and the carrier to reduce the Pt content [130,137]. On the other hand, non-noble metal-based catalysts were developed to replace the noble metal Pt catalysts [138,139,140]. Among them, Ni-based catalysts are the most widely studied [136,141,142,143]. Shukla et al. [137] used metal oxide and perovskite as platinum catalyst carriers and prepared platinum catalysts supported by metal oxide (Pt/MO) via the wet leaching method; the catalysts were tested for the catalytic dehydrogenation of methylcyclohexane. The experimental results showed that the hydrogen evolution rate of Pt/La_2_O_3_ was significantly better than that of the conventional Pt/Al_2_O_3_ catalyst under 3 wt% Pt loading. In an attempt to further improve the catalyst, the author reduced the loading of Pt to 1 wt% and used La_0.7_Y_0.3_NiO_3_ instead of La_2_O_3_ as the catalyst carrier, gaining a hydrogen evolution rate in 90 min twice as high as that of the 3 wt% Pt/La_2_O_3_ catalyst. In addition, no by-product methane was observed in the dehydrogenation of methylcyclohexane, and the dehydrogenation selectivity was close to 100%. In a word, the 1 wt% La_0.7_Y_0.3_NiO_3_ catalyst successfully achieved the goals of achieving low-Pt metal loading as well as high activity and selectivity. 

Pure Ni-based catalysts generally show poor selectivity for the dehydrogenation of naphthenic hydrocarbons [141,144], so the preparation of highly active monometallic nickel-based catalysts is a serious challenge. Gobara et al. [145] successfully prepared a 5%Ni/20%CeO_2_-Al_2_O_3_ catalyst, where CeO_2_-Al_2_O_3_ nanocomposites via the coprecipitation method was used to load metal Ni, whose performance was tested via cyclohexane dehydrogenation. The experimental results showed that when the LHSV of the catalyst is 1 h^−1^, the yield of benzene could reach 99%. This excellent catalytic performance is attributed to the high specific surface area provided by the hierarchical structure support and the reasonable cerium/oxide loading ratio. The catalyst characterization results showed that Ce enhanced the electronic structure of the catalyst and helped produce good Ni dispersion on the surface of the support, thus improving the catalytic activity.

On the one hand, the dehydrogenation reaction of cyclohexane, methylcyclohexane and decalin needed to consume a lot of energy due to their high dehydrogenation temperature (≥200 °C) and reaction enthalpy (63–69 kJ/mol H_2_) [20,146,147]. On the other hand, the hydrogen produced by their dehydrogenation reaction has a high temperature (≥200 °C) and may be accompanied by a certain amount of impurity gas, which is difficult to use directly [67,148,149]. In contrast, N-heterocyclic compounds represented by NEC are more favored, and their dehydrogenation catalysts have also been widely and deeply studied. Table 4 lists the performance of typical catalysts for N-heterocyclic dehydrogenation.

In order to determine the noble metal component with excellent performance, Yang et al. [153] used four conventional noble metal catalysts (5 wt% Ru/Al_2_O_3_, 5 wt% Rh/Al_2_O_3_, 5 wt% Pd/Al_2_O_3_ and 5 wt% Pt/Al_2_O_3_) to catalyze the dehydrogenation of dodecahydro-N-ethylcarbazole (12-NEC). At 180 °C and 101 kPa, the initial catalytic activity follows the order Pd/Al_2_O_3_ > Pt/Al_2_O_3_ > Ru/Al_2_O_3_ > Rh/Al_2_O_3_. Both Pd/Al_2_O_3_ and Pt/Al_2_O_3_ catalysts could achieve complete dehydrogenation, and the selectivity for the product of complete dehydrogenation was up to 100%. Similar results were obtained in the study of Wang et al. [157], where a variety of noble metal (Pd, Pt, Rh, Ru, Au) catalysts using graphene as the catalyst carrier were prepared via a one-pot method to catalyze the dehydrogenation of 12-NEC. Their experimental results showed that the catalytic activity of the catalysts with different noble metals loaded on graphene follows the order: Pd > Pt > Rh > Ru > Au; Pd was still the most active noble metal. However, the experiment of Jiang et al. [159] obtained a different result. They synthesized five noble metal catalysts (Pt/TiO_2_, Pd/TiO_2_, Ru/TiO_2_, Rh/TiO_2_, Au/TiO_2_) with TiO_2_ as the carrier via a sequential reduction method and studied their catalytic performance for the dehydrogenation of 12-NEC. At 453 K and 101 kPa, the dehydrogenation catalytic activity of the noble metal catalysts indicated the following order: Pt/TiO_2_ > Pd/TiO_2_ > Rh/TiO_2_ > Au/TiO_2_ > Ru/TiO_2_. The catalytic activity and selectivity of the Pt/TiO_2_ catalysts were better than that of the Pd/TiO_2_ catalysts. For different catalyst carriers, the comparison results for the catalytic activity of noble metals are not completely the same, but Pt and Pd always show outstanding catalytic activity [153,157,159,160]. Therefore, Pt-based and Pd-based catalysts are also widely used in the dehydrogenation of other N-heterocyclic compounds [161,162,163].

Suitable carriers can form strong interactions with active metal components, thus reducing the amount of metal used. The early carriers of dehydrogenation catalysts for N-heterocyclic compounds mainly include carbon, Al_2_O_3_ and SiO_2_ [153,154,164]. These carriers have the advantages of a wide source, low price and good thermal stability, etc. However, the small specific surface area and the poor dispersion of active metals on them limit their application [74]. Therefore, scholars are committed to finding new suitable carriers for high-performance catalysts. Wang et al. [155] successfully prepared graphene oxide (rGO) and used it as carrier in a 2.5 wt% Pd/rGO-_EG_ catalyst, which exhibited excellent catalytic performance for the dehydrogenation of 12-NEC. The Pd/rGO-_EG_ catalyst had excellent dehydrogenation performance. After 12 h of reaction at 443 K, the conversion of 12H-NEC was 100% and the selectivity of NEC was increased to 84.61%. Compared with the 5 wt% Pd/Al_2_O_3_ commercial catalyst, the specific activity of the 2.5 wt% Pd/rGO-_EG_ catalyst was 14.4 times higher, while the load of Pd was reduced by nearly half. The characterization results showed that the specific monolayer structure of graphene made Pd more evenly distributed on the two-dimensional plane of the graphene sheet, and the two-dimensional plane was also conducive to the full contact between the reaction substrate and the active sites. Gong et al. [156] selected titanium dioxide (TiO_2_) as the catalyst carrier and prepared a series of Pt/TiO_2_ catalysts with different Pt loading via a solvothermal reduction method. These catalysts were tested for the dehydrogenation of 12-NEC under the reaction conditions of 453 K and 101.325kPa. Among the catalysts, 2.5 wt% and 1 wt% Pt/TiO_2_ showed excellent catalytic performance, superior to that of a 5.0 wt% Pt/TiO_2_ catalyst and 5 wt% Pd/Al_2_O_3_ commercial catalyst. Considering the difference in the noble metal loading, the 1 wt% Pt/TiO_2_ catalyst was the best, with its selectivity to NEC up to 98%. XPS results confirmed the existence of oxygen vacancies in the TiO_2_ carrier and strong metal-supporting interaction between Pt and TiO_2_. Additionally, combined with the XRD, HRTEM and TPR results, Gong et al. found that the excellent catalytic performance of the Pt/TiO_2_ catalyst could be attributed to the strong metal–support interaction between the active component Pt and TiO_2_ carriers.

In addition to developing new catalyst carriers, improving conventional catalyst carriers is also a feasible strategy for preparing catalysts with low costs and good performance. Feng et al. [158] selected conventional alumina as the carrier to prepare 1 wt% Pd/Al_2_O_3_ catalysts and used these catalysts for the dehydrogenation of 12H-NPC. They adjusted the morphology, surface acidity and specific surface area of the alumina carrier by controlling the hydrothermal temperature (120–200 °C), thus affecting the dispersion and particle size of precious metal Pd, closely related to the catalyst activity. The experimental results showed that the dehydrogenation catalytic performance of the catalyst with alumina support prepared at 120 °C was the best. After a reaction at 180 °C and 101 kPa for 6 h, the hydrogen evolution amount of 12H-NPC reached the theoretical maximum of 5.43 wt% and the selectivity to the completely hydrogenated product of NPC reached 100 %, which were both significantly better than those of the 5 wt% Pd/Al_2_O_3_ commercial catalyst. Further, the characterization results of the catalyst showed that the Al_2_O_3_ carrier synthesized at 120 °C had weak acidity and the largest specific surface area, which was conducive to the contact of Pd-active sites and reactive components, thus enhancing the catalytic performance. When the catalyst carrier is a single material, there may be shortcomings, such as the low density of active sites and small specific surface area [165,166]. Under such a situation, mixing a second material into the original one to construct the binary composition is an effective means to prepare good catalyst carriers. Yang et al. [150] prepared a series of Pt/SiO_2_-TiO(OH)_2_ catalysts with different Pt loading via the solvothermal method using SiO_2_-TiO(OH)_2_ binary compositions as carriers. The dehydrogenation experiment results of 12-NEC showed that the catalytic performance of 2.5 wt% Pt/SiO_2_-TiO(OH)_2_ was better than that of both the 2.5 wt% Pt/SiO_2_ and 2.5 wt% Pt/TiO_2_(OH)_2_ catalysts, confirming the effectiveness of the binary carrier. Combined with the catalyst characterization method, it was found that the enhanced catalytic activity was due to the interaction between SiO_2_ and TiO_2_(OH)_2_ in the support, which promoted the generation of oxygen vacancies, created more anchor points for the active component platinum particles (Figure 10), and thus enhanced the strong metal-supporting interaction between the metal Pt and SiO_2_-TiO_2_(OH)_2_ support.

#### 2.2.2. Bimetallic Dehydrogenation Catalysts

It is well known that suitable bimetallic catalysts can achieve excellent catalytic performance and low preparation costs. For the dehydrogenation of various LOHCs, a large number of bimetallic catalysts have been developed (Table 5).

The Pt-based catalysts have good catalytic activity and selectivity for the dehydrogenation of naphthenic hydrocarbons; however, they also show poor stability [180,181]. Additionally, during the dehydrogenation of cycloalkanes, coke generated by cracking reactions at acidic sites in the catalyst carriers is deposited on the Pt-based catalyst, resulting in catalyst deactivation [170,180,182,183]. Therefore, researchers added the second metal components, such as Ca [180], Sn [170,184] and Mn [185,186], to improve their stability and promote the activity and selectivity. Nakano [186] examined eleven metals as the second metal component to be added to the commonly used Pt/Al_2_O_3_ catalysts in a search for bimetallic combinations with higher toluene selectivity and better stability in the dehydrogenation of MCH. The characterization results showed that Mn reduced the unsaturated coordination of Pt and thus inhibited the decomposition of toluene and deactivation of the catalyst, although it did not form an alloy with Pt. In order to further enhance the performance of Pt-based catalysts, Yan et al. [170] prepared Pt-Sn/Mg-Al bimetallic catalysts by adding metallic Sn to the Pt catalyst and using Mg-Al mixed metal oxides as the carriers (Figure 11). The results of the dehydrogenation of methylcyclohexane showed that when the content of Sn in the bimetallic catalyst was 0.5 wt%, the Pt-Sn/Mg-Al mixed-metal oxide catalyst had the best catalytic performance. At 300 °C, the conversion of MCH could reach 90%, while the dehydrogenation products were only toluene and hydrogen. Moreover, there was no carbon deposition and conversion reduction after 10 h of catalytic reaction, reflecting the good anti-coking ability and stability of the catalyst. The characterization results show that the electron transfer of Sn leads to the negative charge of Pt, which is beneficial to the adsorption of MCH molecules at Pt-active sites, thus improving the catalytic activity. The good anti-coking ability and stability of the catalysts benefited from the presence of metallic Sn as well as the acid free, pore size Mg-Al mixed-metal oxide carrier.

In order to prepare the dehydrogenation catalysts to be more lower-cost, researchers tried to develop bimetallic catalysts using other cheaper noble or non-noble metals. Among the cheaper precious metals, Ag, which can reduce the degree of hydrogenolysis, seems to be a promising dehydrogenation active ingredient [168]. Pande et al. [80] synthesized stable Ag-Rh bimetallic nanoparticles (BNPs) and prepared several Ag-RH catalysts with different bimetallic ratios using activated carbon cloth (ACC) and Y_2_O_3_ as supports. The results of catalytic dehydrogenation experiments on cyclohexane showed that the 5 wt% Ag_1_-Rh_4_/Y_2_O_3_ catalyst with Y_2_O_3_ as the support and a Ag–Rh ratio of 1:4 exhibited the best catalytic performance. The cyclohexane hydrogen evolution rate could reach 400 mmol/g_met_/min, superior to that with a 5%Pt/ACC catalyst. The addition of Rh, with excellent C-H bond cracking ability, effectively promoted the evolution rate of H_2_ and the conversion rate of benzene, while the Y_2_O_3_ support was helpful for the smooth migration of hydrogen from the active site to the catalyst surface.

In addition, the tantalizing low-cost feature of non-metal-based catalysts attracts some researchers. The catalytic performances of non-noble metal monometallic catalysts are largely unsatisfying; however, the strategy of bimetallic combinations leads to better outcomes [126,144]. Anaam et al. [144] prepared four kinds of bimetallic Ni-M/Al_2_O_3_ catalysts (M = Ag, Zn, Sn, In), using Al_2_O_3_ as a carrier via a homogeneous deposition precipitation method. The catalytic dehydrogenation experiment of methylcyclohexane showed that the Ni-Zn/Al_2_O_3_ catalyst had a higher yield of toluene than the other bimetallic catalysts. Further experimental results showed that the Ni_1_-Zn_0.6_/Al_2_O_3_ catalyst with an optimum Ni–Zn ratio achieved 99% selectivity for toluene at 300 °C—much higher than that of the Ni/Al_2_O_3_ catalyst. The DFT calculation results show that the increase in selectivity is due to the addition of metal Zn, which poisons low-coordinated sites where C–C breaking preferentially occurs. Xia et al. [126] added metal Cu to the Ni/SiO_2_ catalyst that had a poor selectivity of benzene. They prepared the Ni_x_Cu_1−x_/SiO_2_ catalysts (x = 0, 0.8 0.85, 0.9, 1) and used them to catalyze the dehydrogenation reaction of cyclohexane to evaluate their catalytic performance. The Ni_0.85_Cu_0.15_/SiO_2_ catalyst showed the best catalytic performance, with 94.9% cyclohexane conversion and 99.5% benzene selectivity at 350 °C. The addition of an appropriate amount of Cu significantly increased the selectivity for benzene over Ni/SiO_2_ catalysts from about 40% to 99%. 

For N-heterocyclic compound systems represented by ethylcarbazole, both Pt and Pd exhibit good dehydrogenation activities, but the latter is cheaper and also easier to form alloys with other metals [187]. Therefore, the development of bimetallic dehydrogenation catalysts for N-heterocyclic compounds has been mostly focused on Pd-based catalysts, and a suitable ratio of bimetallic composition can improve the catalyst activity while reducing the Pd content. Wang et al. [178] prepared a series of Pd-Cu catalysts with different bimetallic ratios via the one-pot method, in which reduced graphene oxide (rGO) was adopted as the carrier. The performance of each catalyst was evaluated using the dehydrogenation reaction of 12-NEC, and the experimental results were shown in Figure 12. Pd_1.2_Cu/rGO and Pd_1.8_Cu/rGO catalysts both showed good catalytic performance. Considering the cost of catalysts, Pd_1.2_Cu/rGO is the ideal one. Compared with the Pd/rGO catalyst, the Pd_1.2_Cu/rGO catalyst reduced the load of Pd by 20% while maintaining the same catalytic activity and selectivity, which undoubtedly reflected the superiority of the bimetallic catalyst. The excellent performance of the Pd_1.2_Cu/rGO catalyst originated from the smaller metal particle size as well as the similar electronic structure to pure metal Pd. Chen et al. [172] synthesized the Pd-Ni/Al_2_O_3_ catalyst with a bimetallic ratio of 1:1 via the wet impregnation method and used it to catalyze the dehydrogenation reaction of perhydro-N-propylcarbazole(12-NPC). Compared with the Pd/Al_2_O_3_ catalyst with similar metal loading, the Pd-Ni/Al_2_O_3_ catalyst with a reduction of nearly 50% in terms of Pd metal loading had better catalytic activity. In addition, the Pd-Ni/Al_2_O_3_ catalyst had good stability and its catalytic activity had no degradation after five reactions at 200 °C.

In order to obtain a better bimetallic combination, it is necessary to expand the range of metals and conduct systematic screening. Kustov et al. [187] incorporated the second metal component (Ru, Pt, Cr, Ni, Ge and W) to the Pd/TiO_2_ catalyst and found that for the dehydrogenation of 12-NEC, bimetallic Pd-Ru/TiO_2_ and Pd-Pt/TiO_2_ (the second metal content was 10 at%) catalysts possessed better catalytic activity and selectivity than Pd/TiO_2_ catalysts. Wang et al. [175] prepared a bimetallic Pd–M/rGO catalyst by combining four metals—Au, Ag, Ru and Rh—with metal Pd using reduced graphene oxide as a carrier. For different combinations of bimetallic catalysts with the same molar ratio, the experimental results of the catalytic dehydrogenation of 12-NEC revealed the catalytic activity following the order of Au_1_Pd_1.3_ > Au_1_Pd_2_ > Au1Pd_1_ > Ru_1_Pd_1.3_ > Au_1_Pd_0.7_ > Rh_1_Pd_1.3_ > Ag_1_Pd_1.3_ (Figure 13a). Among the Pd-Au/rGO catalysts containing the best bimetallic combination, the Au_1_Pd_1.3_/rGO catalyst showed the most ideal performance, with nearly 100% conversion of 12-NEC and 100% selectivity of NEC within 4 h. In addition, the Au_1_Pd_1.3_/rGO catalyst showed excellent stability, with a 100% conversion of 12-NEC and 98% selectivity of NEC after five catalytic reactions, as shown in Figure 13b. The research team of Fang et al. successively prepared a series of bimetallic Pd-M/SiO_2_ (M = Au [174], Ni [173], Cu [173]) catalysts for the dehydrogenation of 12-NEC. Their experimental results show that the optimum molar ratio of Pd to the second active metal in the bimetallic Pd–M/SiO_2_ catalyst is 3:1, gaining a significantly better catalytic performance than the Pd/SiO2 monometallic catalyst. After 8 h of a reaction at 453 K and 101.3 kPa, 5.7 wt% hydrogen evolution and 94.9% selectivity of NEC were achieved with the catalysis of Pd_3_-Au_1_/SiO_2_. Characterization results of the catalysts revealed that the dehydrogenation activity and selectivity of bimetallic catalysts depend on the components of the bimetallic catalyst and the degree of alloying.

Apart from appropriate components and proportions, the use of carriers with excellent properties is another effective way to develop highly active bimetallic catalysts. Feng et al. [177] prepared a series of PdxNiy/KIT-6 catalysts using an ordered mesoporous SiO_2_ material, KIT-6, to support metal Pd and Ni, and used these catalysts to catalyze the dehydrogenation of 12-NEC. Pd_4_Ni_1_/KIT-6 with a mass ratio for Pd to Ni of 4:1 had the best catalytic performance. Under its action, the dehydrogenation efficiency of 12-NEC after a reaction at 180 °C for 6 h was up to 99.1%, partly owing to the ordered mesoporous structure and large surface area in the KIT-6 carrier. Tang et al. [172] successfully prepared a bimetallic Pd-Co/NGC catalyst with magnetic n-doped partially graphitized carbon(NGC) as carrier, obtaining good catalytic performance for the dehydrogenation of 12-NEC. The PdCo/NGC catalyst had excellent catalytic performance for the dehydrogenation of 12-NEC. After a reaction at 180 °C for 6 h, the conversion of 12H-NEC was 100%, and the selectivity of NEC was 97.87%. The excellent catalytic performance of the PdCo/NGC catalyst was partly due to the doping of N in the catalyst carrier and the large surface area of the carrier itself. The former enhanced the interaction between the active components of PdCo and the carrier, while the latter was conducive to the dispersion of PdCo particles.

## 3. Bifunctional Catalysts of LOHCs

For a liquid organic hydrogen carrier system, different catalysts are usually utilized in the hydrogenation and dehydrogenation process, making the whole LOHC system more complex and costly, particularly in the application of stationary hydrogen storage. The bifunctional catalysts can realize the efficient hydrogenation and dehydrogenation reactions of LOHCs in one reactor. This will promote the heat coupling of the processes of hydrogenation and dehydrogenation and improve the economy of LOHC systems [76,188]. The catalytic performance of the bifunctional catalyst can be measured using the combination (Figure 14) of a high-pressure autoclave, high-precision pressure gauge and gas flowmeter [68,189]. Taking the NEC system as an example, in the process of hydrogenation, NEC and the catalyst are loaded into the autoclave. Then, the autoclave is heated to the specified temperature and filled with quantitative hydrogen. The hydrogenation rate is measured and calculated by monitoring the change in pressure in the system (Sievert method) [190]. Normally, the hydrogenation reaction is complete when the system pressure remains stable for more than 2 h [68]. After the hydrogenation reaction is completed, the obtained 12H-NEC can be used for the dehydrogenation reaction. The remaining H_2_ after the hydrogenation experiment is released and vacuumed from the reaction system. Then, the designated hydrogen pressure is filled into the autoclave and heat autoclave to the specified temperature. The flow rate of hydrogen released is recorded using the flow meter and the hydrogen release is calculated by integrating the instantaneous flow recorded [190]. The final H_2_ uptake and release amount was analyzed and corrected using the NMR or GC-MS results of the samples of the products after hydrogenation and dehydrogenation [190,191]. Table 6 summarizes the performance of typical bifunctional catalysts of LOHCs.

The carriers with functional properties (CeO_2_, LaNi_5_…) can contribute to the realization of the highly efficient bifunctionality of catalysts. For example, CeO_2_, characterized by excellent chemical stability as well as the ability to form oxygen vacancies, has received extensive attention [195,196]. Zhou et al. [197] took different methods to prepare Pd/CeO_2_ catalysts with different CeO_2_ structures using CeO_2_ as a carrier and evaluated the performance of these catalysts via benzene–cyclohexane conversion reactions. Experimental results showed that all Pd/CeO_2_ catalysts maintained 100% selectivity for the reaction products cyclohexane/benzene. Among them, the Pd/CeO_2_-HT catalyst with an ordered mesoporous structure and large specific surface area beneficial to the adsorption and diffusion of reactants had the best catalytic performance. During hydrogenation, a H_2_ molecule is adsorbed and activated by metal Pd, and a benzene molecule is activated by an oxygen vacancy formed by oxygen reduction on the surface of the CeO_2_ carrier. Afterwards, the activated hydrogen on the surface of the palladium metal will overflow to the surface of the reduced CeO_2_ carrier, thus realizing the hydrogenation of benzene. In the dehydrogenation process, the presence of metallic Pd exhibits strong ability to cleave C-H bonds, and hence the H atoms generated are strongly adsorbed on the surface of Pd and combined to form H_2_ molecules. At an appropriate temperature, H_2_ molecules desorb from the surface of Pd, thus achieving cyclohexane dehydrogenation. Chen et al. [198] also selected CeO_2_ as the carrier. They ingeniously used a Pt atom to replace a Ce atom on the surface of CeO_2_ and the prepared the Pt1/CeO_2_ single platinum catalyst so it could catalyze the hydrogenation and dehydrogenation of toluene/methylcyclohexane and benzene/cyclohexane systems. Pt_1_/CeO_2_ had excellent dehydrogenation activity. The dehydrogenation rate of cyclohexane and methylcyclohexane can reach 32,000 mol H_2_ mol Pt^−1^ h^−1^ with its catalysis, which is 309 times of that of a conventional Pt/Al_2_O_3_ catalyst under the same reaction conditions. Under low oxygen partial pressure and high temperature conditions, Ce^4+^ in CeO_2_ is easily converted into Ce^3+^, which releases oxygen ions in the lattice and generates oxygen vacancies [199]. The characterization results of The Ce 3d XPS spectroscopy showed that there is a high concentration of Ce^3+^ in the CeO_2_ carrier of the Pt_1_/CeO_2_ catalyst, proving the existence of oxygen vacancies. The excellent catalytic activity of the Pt_1_/CeO_2_ catalyst was derived from the strong synergistic effect existing between the Pt atom, the rich oxygen vacancies in the support and the variable redox of CeO_2_. The existence of surface oxygen vacancies provides a channel for the combination of reactants and Pt sites, making it easier for the adsorption of reactants. The redox coupling between Pt and Ce ions facilitates H abstraction and H species spillover, and the reducible ceria carrier acting as a reservoir for abstracted hydrogen is conducive to the continuous dehydrogenation of adsorbed reactants and the release of H_2_.

In addition to CeO_2_, the hydrogen storage alloy materials that can rapidly absorb and desorb hydrogen are also ideal carriers for bifunctional catalysts. Yu et al. [193] innovatively used LaNi_5_, which is one of the most commonly used hydrogen storage alloys [200], as the catalyst for reversible hydrogen storage of NEC. They prepared LaNi_5+x_ (x = 0, 0.5, 1) particles with a size of about 100 nm using the improved CaH_2_ reduction method in KCl molten salt. The excess Ni and LaNi_5_ played the roles of the metal active component and catalyst carrier, respectively. The results of the catalytic reaction of ethyl carbazole showed that LaNi_5.5_ had the best comprehensive catalytic performance, with which 5.5 wt% hydrogen storage could be achieved via the hydrogenation of NEC at 453 K for 4.5 h, and 5.5 wt% hydrogen release could be achieved via the dehydrogenation of 12H-NEC at 473 K for 4.0 h. The good catalytic activity of the LaNi_5.5_ catalyst was partly due to the synergism between excessive Ni and LaNi_5_. The TM/MH (Ni/LaNi_5_) interface formed by excess Ni and LaNi_5_ could effectively promote the H transfer between the H atoms in the LaNi_5_-H solid solution and the activated NEC/12H-NEC molecules on Ni. Moreover, the existence of LaNi_5_ with good hydrogen absorption/desorption kinetics accelerates the H transfer between H_2_ and atomic H. The bi-directional catalytic ability of the LaNi_5.5_ catalyst is due to the existence of the LaNi_5−H_ solid solution. The pressure-dependent H concentration of the LaNi_5-H_ solid solution is critical to drive H transfer in opposite directions. Then, Yu et al. [190] prepared a Pd/LaNi_5_ catalyst by depositing metal Pd on the surface of LaNi_5_ particles. The 1 wt% Pd/LaNi_5_ catalyst has the best bidirectional catalytic performance for NEC/12H-NEC system among the literature collected. Under its catalysis, NEC could complete 5.5 wt% H_2_ absorption in 0.7 h, and 12-NEC could complete 5.5 wt% H_2_ release in 2.1 h. The hydrogenation/dehydrogenation process of NEC/12-HNEC on Pd/LaNi_5_ and Pd/Al_2_O_3_ is shown in Figure 15. In the hydrogenation process, LaNi_5_ can rapidly absorb H_2_ molecules and convert them into atomic hydrogen. Atomic H can be stored in the lattices of LaNi_5_ and then made to react with nH-NEC (*n* = 0, 4, 8, 12) molecules adsorbed on the surface of Pd via permeation through the Pd layer. In addition, atom H released by the nH-NEC molecule can also be stored in LaNi_5_ lattices and then desorbs during the hydrogenation process. In general, LaNi_5_ support in Pd/LaNi_5_ provides a new hydrogen transfer pathway compared with a conventional Pd/Al_2_O_3_ catalyst. The abundant H-binding sites in LaNi_5_ breaks the limitation of an insufficient number of accommodating bond sites of atom H on conventional catalysts, promoting hydrogenation and dehydrogenation reactions. Moreover, the rapid H diffusion kinetics of LaNi_5_ and the stable Pd/LaNi_5_ interface also contribute to the realization of excellent catalytic performance.

In addition to the use of carriers with special properties, adding promotive substances to the original catalyst is also an effective method to prepare efficient and bifunctional catalysts. Li et al. [192] used Ce metal to promote a Pd/Al_2_O_3_ catalyst. They synthesized a CeO_2_-Al_2_O_3_ composite carrier with a different CeO_2_ content via a coprecipitation method and prepared Pd/CeO_2_-Al_2_O_3_ catalysts. Pd/CeO_2_-Al_2_O_3_ with a theoretical mass percentage of 20% CeO_2_ (Pd/CeAl-20) had the best performance in the reversible hydrogenation and dehydrogenation of NPCZ. The characterization results of the catalyst showed that the addition of CeO_2_ not only weakened the strong interaction between Pd particles and the support, but also promoted the dispersion of Pd particles on the surface of the carrier, hence providing more active sites. Additionally, the Pd-O-Ce structure formed by Pd and CeO_2_ can effectively regulate the electronic state of Pd, thus enhancing the catalytic performance. Wu et al. [68] used a nonstoichiometric Yttrium hydride (YH_3−x_) to promote a Co-B/Al_2_O_3_ catalyst and obtained exciting results. They prepared the Co-B/Al_2_O_3_-YH_3−x_ catalyst by grinding Co-B/Al_2_O_3_ and YH_3−x_, showing excellent catalytic performance for the hydrogenation and dehydrogenation of the NEC system. In the Co-B/Al_2_O_3_-YH_3−x_ catalyst, the role of Co-B is to activate NEC/12H-NEC molecules, while the role of YH_3−x_ is to transfer H rapidly. The lattice H and H vacancy in YH_3−x_ efficiently promoted the rapid transfer of hydrogen. The specific hydrogen transfer mechanism is shown in Figure 16. The H_2_ pressure controlled the H chemical potential of non-stoichiometric YH_3−x_, thus realizing spontaneous two-way H transfer. The activated NEC and 12H-NEC molecules conduct H transfer through the Co-B/YH_3−x_ interface.

The order of hydrogenation and dehydrogenation catalytic activity of several noble metals is different. For the hydrogenation reaction of LOHCs, Ru has good catalytic activity, while Pd and Pt have good dehydrogenation catalytic activity. Therefore, the combination of one metal with good hydrogenation activity and another metal with good dehydrogenation activity seems a potential way to develop efficient bifunctional catalysts intuitively, which is confirmed by the research results of some scholars. Forberg et al. [194] used silicon carbide nitride (SiCN) as a catalyst carrier to load noble metals Ru and Pd, and they screened the best catalyst, Pd_2_Ru@SiCN, through experiments. For the reversible reaction system of NEC, the Pd@SiCN catalyst only showed good dehydrogenation activity, the Ru@SiCN catalyst showed good hydrogenation activity and extremely weak dehydrogenation activity, while the Pd_2_Ru@SiCN catalyst had both hydrogenation and dehydrogenation activities at the same time. Zhu et al. [67] also reported a bifunctional catalyst Ru-Pd/Al_2_O_3_ for the hydrogenation of NPC and dehydrogenation of 12H-NPC. The Ru-Pd/Al_2_O_3_ catalyst demonstrated superior hydrogenation and dehydrogenation catalytic activity for Ru/Al_2_O_3_ and Pd/Al_2_O_3_ catalysts, respectively. Xue et al. [191] prepared a bimetallic Pd-Rh nanoparticles (NPs) catalyst, which had good catalytic activity for the hydrogenation and dehydrogenation of the NEC system. The 0.6%Rh–1% Pd/γ-Al_2_O_3_ (Pd_4_Rh_2_/γ-Al_2_O_3_) catalyst had the best comprehensive performance, while the corresponding monometallic catalyst only showed good activity in either hydrogenation or dehydrogenation. DFT calculation results showed that the hydrogenation process from NEC to 4H-NEC and 6H-NEC to 8H-NEC was exothermic; in addition, they also showed that the dehydrogenation process from 12H-NEC to 8H-NEC and 6H-NEC to 4H-NEC on the Pd_4_Rh_2_/γ-Al_2_O_3_ easily occurred, evidencing the reversible catalysis process for hydrogenation and dehydrogenation.

## 4. Conclusions and Outlook

LOHCs are thought of as a promising for the large-scale, long distance transport of hydrogen energy, where catalysts for the hydrogenation and dehydrogenation of LOHCs play a crucial role. In this review, we introduce a series of catalytic systems from single-function catalysts to bifunctional catalysts and point out the optimization strategy of catalyst performance and past development directions. It was concluded that the properties of carriers, the interaction between metals and carriers, the electronic structure and dispersion of active metals and the positive synergy between multiple metals could significantly affect the final catalytic performance of the catalysts. At present, the development of catalysts with LOHCs with excellent performance still faces some challenges. Advanced noble metal-based catalysts show encouraging catalytic performance in mild conditions. However, the high cost and shortage of precious metals limit their large-scale application. Non-precious metal catalysts with good activity and selectivity are expected to replace noble metal catalysts. However, compared with noble metal catalysts, the performance of non-noble metal catalysts still has great room for improvement. In addition, most of the excellent bimetallic combinations in bimetallic catalysts are screened through pure trial and error, lacking the understanding of the catalytic properties of metal atoms themselves and the guidance of corresponding theoretical calculation results. In view of the above problems, there are several ways to develop excellent performance catalysts of LOHCs in the future.

(1)Compared with monometallic catalysts, catalysts with an appropriate bimetallic combination and proportion have achieved better catalytic performance. On this basis, we can try to develop tri-metal or even high-entropy alloy catalysts to achieve the goals of better economic efficiency and catalytic performance. Researchers need to explore different metal compositions, screen out excellent metal combinations and proportions and construct a positive synergy between multiple metals.(2)It is an effective way to develop efficient and multifunctional catalysts by using catalyst carriers with excellent characteristics (such as rapid hydrogen transfer capability or the activation ability of hydrogen or LOHCs). Additionally, improving existing, commonly used carriers can also effectively improve catalyst performance. In this respect, using defect engineering to construct carrier surface defects to anchor active metals or adding promotive compounds (such as CeO_2_) to enhance the interaction between active metals and carriers have been proven to be effective methods for improving carriers.(3)It is necessary to develop hydrogenation and dehydrogenation bifunctional catalysts to satisfy the application requirements of fuel cells, renewable energy storage systems and other fields. The bifunctional catalysts can effectively improve the efficiency and economy of the entire system, representing future development requirements.(4)The reaction mechanism of LOHCs, the reaction pathway on different catalysts and the catalytic properties of noble metals can be further revealed using theoretical calculation. The theoretical calculation results can provide a theoretical guide for the combination of multiple metals, the optimization of ratio and the selection of excellent carriers.

## Figures and Tables

**Figure 1 materials-16-03735-f001:**
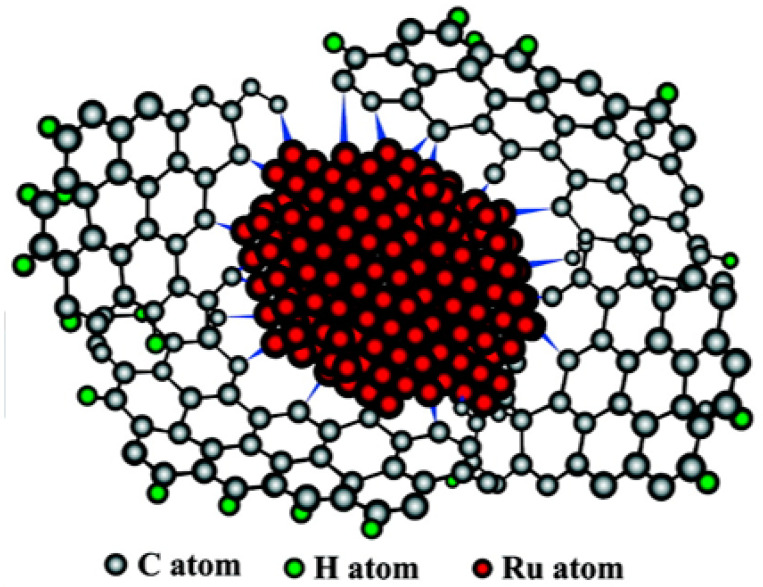
Ru−C linkages formed between a Ru nanoparticle and the edge of the graphene (blue arrows) (adapted with permission from [93]. Copyright {2008} American Chemical Society).

**Figure 2 materials-16-03735-f002:**
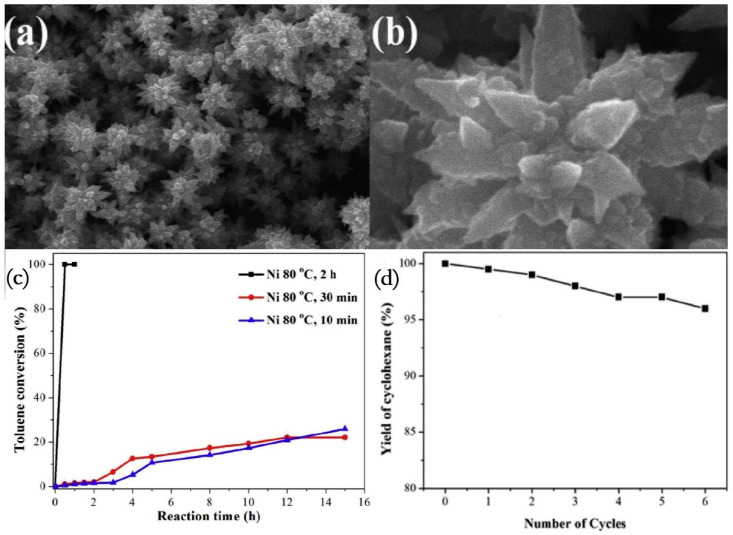
(**a**,**b**) SEM images of as-synthesized Ni nanocrystals prepared at 80 °C for 2 h. (**c**) Results of the catalytic hydrogenation of toluene over Ni nanocrystals prepared at 80 °C for different growth times, and (**d**) the catalytic performance of recycling Ni nanocrystals prepared at 80 °C for 2 h [96].

**Figure 3 materials-16-03735-f003:**
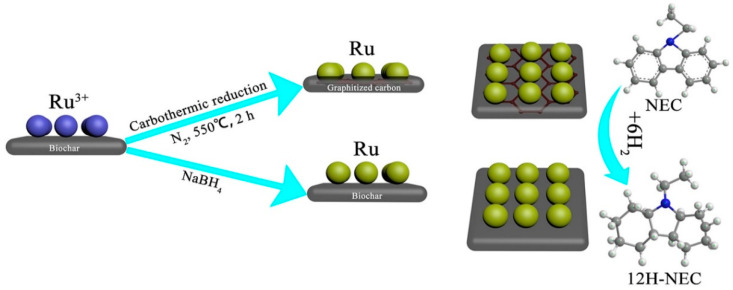
The proposed schematic diagram of the Ru/pg-BC catalyst prepared via a carbothermal reduction [100].

**Figure 4 materials-16-03735-f004:**
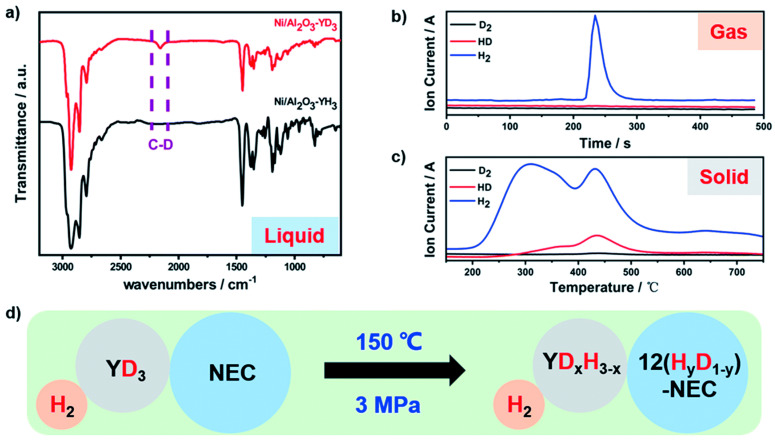
(**a**–**c**) Analysis of the D distribution in the liquid, gas and solid phases after hydrogenation of NEC using Ni/Al_2_O_3_−YD_3_ in a sealed autoclave (150 °C, 3 MPa H_2_). (**a**) FT−IR spectrum of the liquid product. The D-free 12H–NEC spectrum is shown for comparison. (**b**) The mass spectrum of the gas phase. (**c**) Mass spectrum of the released gas when the solid catalyst was heated with Ar flow (10 °C min^−1^ to 800 °C). (**d**) Schematic illustration of the isotope distribution before and after hydrogenation [97].

**Figure 5 materials-16-03735-f005:**
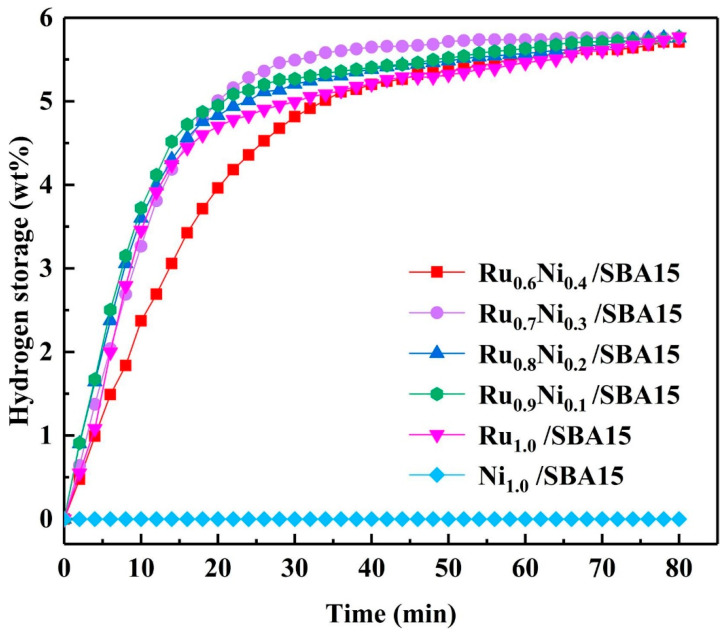
Different Ru/Ni ratio of Ru_x_Ni_1−x_/SBA15 hydrogenation curves [120].

**Figure 6 materials-16-03735-f006:**
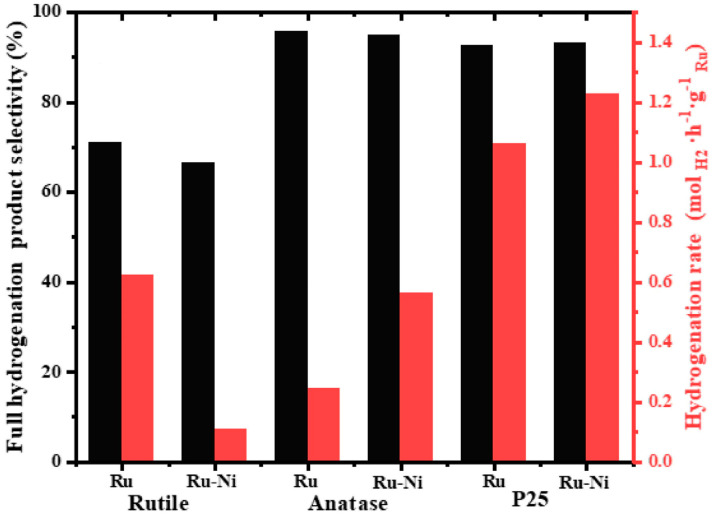
Catalytic performances of as-prepared catalysts. The left axis shows selectivity toward full hydrogenation product 12H-NEC (after reacting 24 h); the right axis shows hydrogenation rate. Reaction condition: 150 °C, 70 bar H_2_, 2.5 g NEC, 0.125 g catalyst, 24 h, stirring at 200 rpm [105].

**Figure 7 materials-16-03735-f007:**
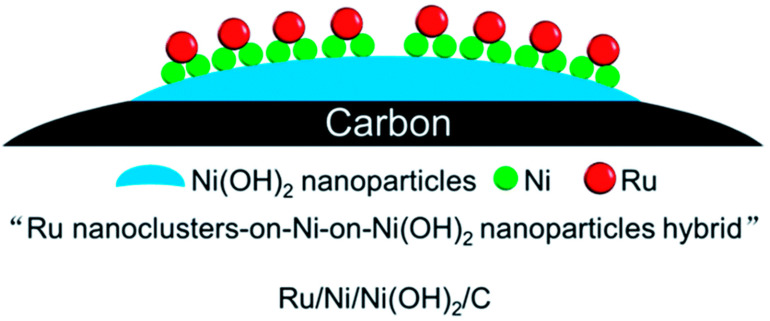
Design of the ruthenium nanoclusters-on-nickel-on-nickel hydroxide nanoparticle hybrid (Ru/Ni/Ni(OH)_2_/C) [116].

**Figure 8 materials-16-03735-f008:**
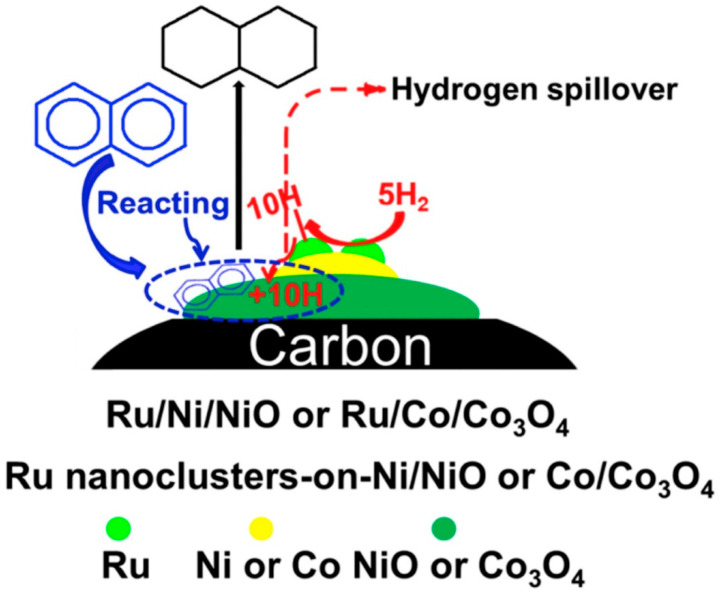
Mechanisms for naphthalene hydrogenation to decalin over the Ru/Ni/NiO/C or Ru/Co/Co_3_O_4_/C catalysts [117].

**Figure 9 materials-16-03735-f009:**
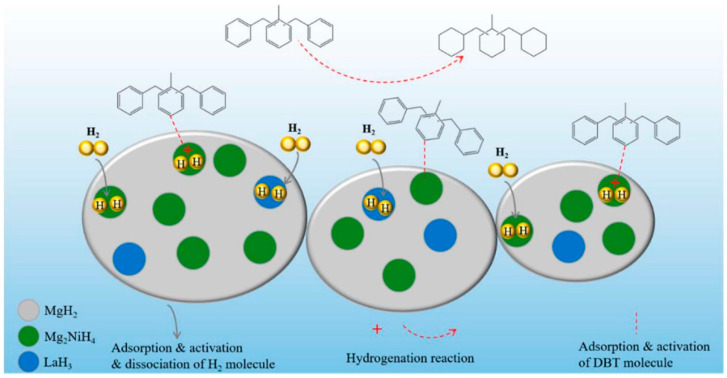
Scheme of the proposed reaction mechanism for the hydrogenation process of dibenzyltoluene catalyzed by Mg-based metal hydrides [124].

**Figure 10 materials-16-03735-f010:**
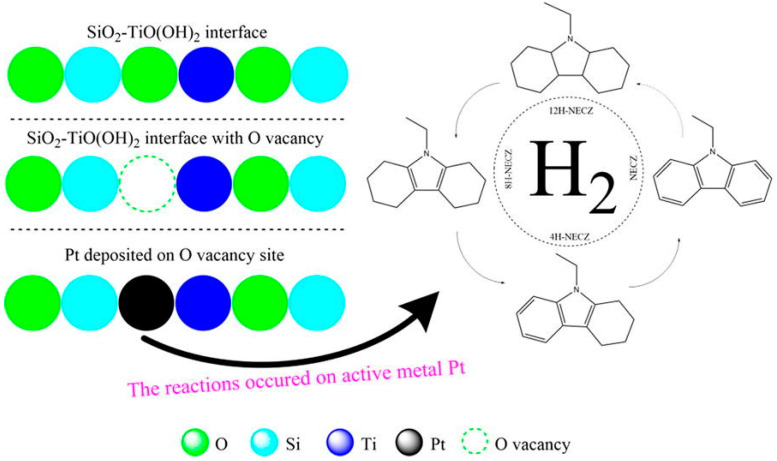
Schematic diagram of 12-NEC dehydrogenation catalyzed by the Pt/SiO_2_-TiO(OH)_2_ catalyst [150].

**Figure 11 materials-16-03735-f011:**
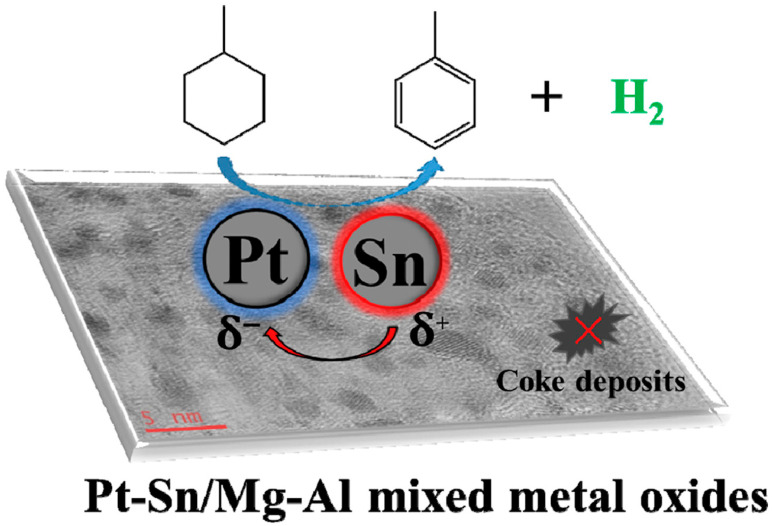
Schematic diagram of methylcyclohexane dehydrogenation catalyzed by the Pt-Sn/Mg-Al mixed metal oxide catalyst [170].

**Figure 12 materials-16-03735-f012:**
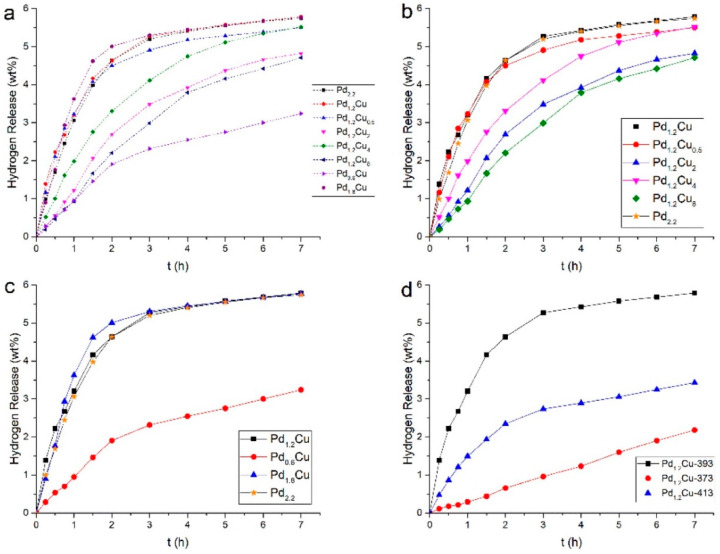
Dehydrogenation efficiency of different catalysts at 453 K over time. (**a**) global comparison, (**b**) specific performance comparison using the changing amount of Cu, (**c**) specific performance comparison by changing the amount of Pd, (**d**) specific performance comparison by changing the reduction temperature in the co-reduction process [178].

**Figure 13 materials-16-03735-f013:**
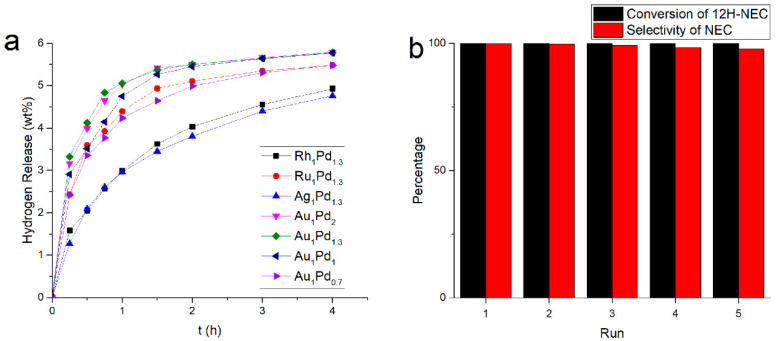
(**a**) Hydrogen release amount for 12H-NEC on different catalysts over time at 453 K. (**b**) Cycle performance of Au_1_Pd_1.3_/rGO at 453 K (reprinted with permission from [175]. Copyright {2019} American Chemical Society).

**Figure 14 materials-16-03735-f014:**
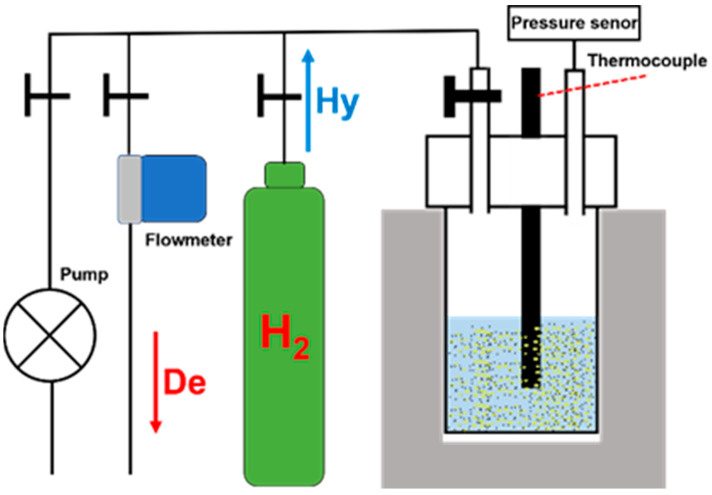
Schematic diagram of the measurement setup [this figure has been published in CCS Chemistry [2020]] [Nonstoichiometric Yttrium Hydride–Promoted Reversible Hydrogen Storage in a Liquid Organic Hydrogen Carrier] are available online at [DOI: 10.31635/ccschem.020.202000255] [68].

**Figure 15 materials-16-03735-f015:**
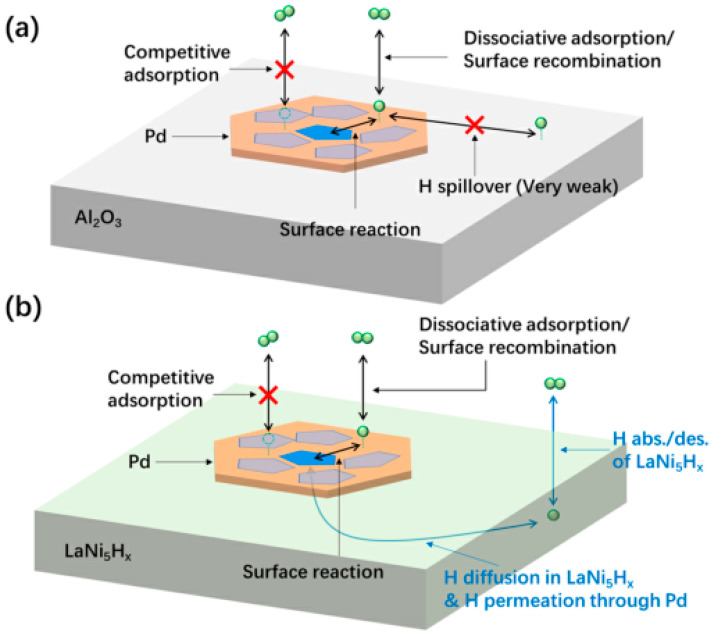
Schematic illustration of the catalytic hydrogenation/dehydrogenation processes on (**a**) Pd/Al_2_O_3_ and (**b**) Pd/LaNi_5_ [190].

**Figure 16 materials-16-03735-f016:**
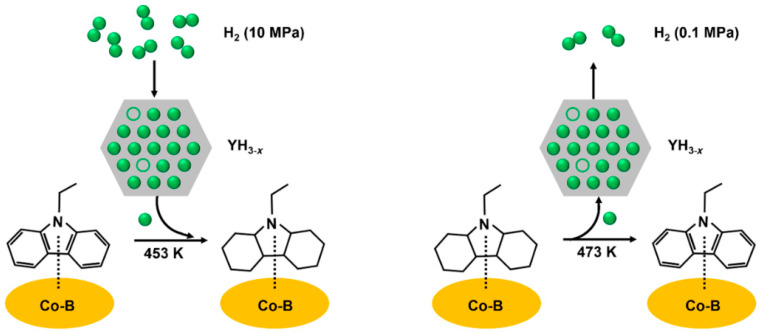
The YH_3−x_-mediated H transfer mechanisms in NEC hydrogenation and 12H-NEC dehydrogenation reactions of the Co-B/Al_2_O_3_-YH_3−x_ catalyst [this figure has been published in CCS Chemistry [2020]] [Nonstoichiometric Yttrium Hydride–Promoted Reversible Hydrogen Storage in a Liquid Organic Hydrogen Carrier] are available online at [DOI: 10.31635/ccschem.020.202000255] [68].

**Table 1 materials-16-03735-t001:** Physical parameters and hydrogen storage capacity of some LOHCs (hydrogen-deficient carriers).

LOHCs	Melting Point (°C)	Boiling Point (°C)	H_2_ Storage Capacity wt%	Reference
Benzene (Ben)	5.5	80	7.2	[20]
Toluene (TOL)	−95	111	6.2	[20]
Naphthalene	80	218	7.3	[20]
Carbazole	245	355	6.7	[20]
N-ethylcarbazole (NEC)	69	378	5.8	[20]
N-propylcarbazole (NPC)	48	336	5.43	[55]
Dibenzyltoluene (DBT)	−39~−34	390	6.2	[38]
1-methylindole (1-MID)	−20	239	5.76	[57]
2-methylindole (2-MID)	57	273	5.76	[58]
1,2-dimethylindole (1,2-DMID)	55	260	5.23	[10]
N-ethylindole (NEID)	−17.8	253.5	5.23	[59]
7-ethylindole (7-EID)	14	230	5.23	[61]
2-(N-methylbenzyl)-pyridine (MBP)	−50.1~−40.2	291~293	6.15	[62]
Acridine (ACD)	111	253	7.25	[47]

**Table 2 materials-16-03735-t002:** Hydrogenation catalytic performance of some catalysts for LOHCs.

LOHCs	Catalysts	T (°C)	P_H2_ (MPa)	Time (h)	Conv ^a^. (%)	Yield ^b^ (%)	TOF (h^−1^)	Ref.
BEN	Ru/SBA-15	20	1	-	100	100	85.3	[72]
BEN	Ru/MOF	60	6	1.5	100	100	3200	[92]
BEN	4.2 wt% Ru/C-silica	110	8	0.53	100	99.8	37,700	[93]
BEN	Ru(0)-Zeolite-Y	22	0.28	1	100	100	1040	[94]
BEN	Ru/CNTs	80	4	0.5	100	99.97	6983	[95]
BEN	Pd/SiO_2_ (co-SEA)	150	7	6	84.1	84.1	-	[91]
TOL	Ni nanoflowers	140	5	0.5	100	100	-	[96]
TOL	Pd/SiO_2_ (co-SEA)	150	7	6	85.4	85.4	-	[91]
TOL	Pt (MP)/CeO_2_-A-400	100	0.5	3	90.8	90.8	-	[77]
NAP	Pt/WO_3_-500	70	3	1	100	100	-	[89]
NAP	Pd/HY-9.5	200	4	1	100	73.15	-	[90]
TEN	1 wt% Ni/Al_2_O_3_–YH_3_	150	10	5	-	95	-	[97]
DBT	Ni70/AlSiO-1/1	150	7	1.5	100	100	-	[98]
DBT	0.3 wt% Pt/Al_2_O_3_	270	3	1.42	-	100	--	[76]
DBT	5 wt% Pd/Al_2_O_3_	260	3	6	-	100	-	[99]
NEC	Ru/pg-BC	130	6	1.17	100	99.41	-	[100]
NEC	Raney-Ni	180	5	1.3	-	86.2		[101]
NEC	Ni70/AlSiO-1/1	150	7	1.5	100	100	--	[98]
NEC	1.3 wt% Ru/YH_3_	130	7	2.5	100	100	-	[102]
NEC	5 wt% Ru/TiO_2_	130	7	-	-	95	-	[103]
NEC	Ru black	130	7	-	-	85	-	[103]
NEC	1.5 wt% Ru-Ni_1_Al_2_-LDO	150	8	1	100	100	-	[104]
NEC	1wt%Ni/Al_2_O_3_–YH_3_	180	10	1.5	100	100	-	[97]
NEC	5 wt% Ru/LDH-3.9CNT	120	6	0.4	100	98.31	-	[82]
NEC	Ru/P25	150	7	24	100	92.4	-	[105]
NEC	Ru/anatase	150	7	24	100	95.7	-	[105]
NEC	Ru/Ni-Fe LDH	110	6	1.33	-	98.88	-	[83]
NPC	5 wt% Ru/Al_2_O_3_	150	7	0.5	-	100	-	[55]
NPC	Ni70/AlSiO-1/1	150	7	1	100	100		[98]

^a^: Conversion of LOHCs (hydrogen-deficient carriers); ^b^: yield of complete hydrogenation products.

**Table 3 materials-16-03735-t003:** The hydrogenation performance of typical bimetallic catalysts of LOHCs.

LOHCs	Catalysts	T (°C)	P (MPa)	Time (h)	Conv ^a^. (%)	Yield ^b^ (%)	Ref.
BEN	Pd-Ni/SiO_2_ (co-SEA)	150	7	6	99.9	99.9	[91]
BEN	Pd-Pt/SiO_2_ (co-SEA)	150	7	6	90.8	90.8	[91]
BEN	0.024 wt% Ru–1.00 wt% Ni/C	60	700 psi	2	100	100	[110]
BEN	Ru_0.56_Ni_0.44_/C	60	5.3	0.5	-	99.8	[113]
BEN	1 wt% Ru_2_Pt_1_ MIL-101	60	1	6	100	100	[85]
TOL	Pd-Ni/SiO_2_ (co-SEA)	150	7	6	99.9	99.9	[91]
TOL	Pd-Pt/SiO_2_ (co-SEA)	150	7	6	91.4	91.4	[91]
TOL	6 wt% Pt_1_Pd_1_/HBEA	150	7–12	2	100	100	[114]
TOL	Pt–Rh/MWNTs	20	1	3	100	100	[115]
NAP	Ru/Ni/Ni(OH)_2_/C	100	4.48	1	-	>99	[116]
NAP	Ru/Ni/NiO/C	100	4.5	1	100	100	[117]
NAP	Ru/Co/Co_3_O_4_/C	100	4.5	0.8	100	100	[117]
NEC	5.0 wt% Ni_0.5_Ru_4.5_/pg-BC	130	6	1.17	100	99.06	[118]
NEC	5.0 wt% Co@Ru/NGC	130	6	1	100	99.1	[119]
NEC	Ru-Ni/P25	150	7	24	100	93	[105]
NEC	Ru-Ni/anatase	150	7	24	100	94.8	[105]
NEC	Ru_0.7_Ni_0.3_/SBA15	100	5	1.33	100	99.82	[120]
NPC	Ru_2.5_Ni_2.5_/Al_2_O_3_	150	4	0.5	100	100	[121]

^a^: Conversion of LOHCs (hydrogen deficient carriers); ^b^: yield of complete hydrogenation products.

**Table 4 materials-16-03735-t004:** The performance of typical catalysts for N-heterocyclic dehydrogenation.

LOHCs	Catalysts	T (°C)	P (MPa)	Time (h)	Conv ^a^. (%)	Yield ^b^ (%)	H_2_ Release (wt%)	Ref.
12H-NEC	5 wt% Pd/NGC	180	0.1	10	100	98.72	5.76	[75]
12H-NEC	2.5 wt% Pt/SiO_2_-TiO(OH)_2_	180	0.1	7	100	97.9	5.75	[150]
12H-NEC	2.5 wt% Pd/LDHs-us	180	0.1	6	100	-	5.72	[151]
12H-NEC	1 wt% Pd-EU/KIT-6	190	0.1	6	100	100	-	[152]
12H-NEC	5 wt% Pd/Al_2_O_3_	180	0.1	4	100	100	-	[153]
12H-NEC	5 wt% Pt/Al_2_O_3_	180	0.1	5	100	100	-	[153]
12H-NEC	4 wt% Pd/SiO_2_	170		1.6	100	100	5.8	[154]
12H-NEC	2.5 wt% Pd/rGO-_EG_	170	0.1	12	100	84.61	5.49	[155]
12H-NEC	5 wt% Pt/TiO_2_	180	0.1	6	100	79	5.38	[156]
12H-NEC	2.32 wt% Pd/rGO	180	0.1	-	100	97.65	5.74	[157]
12H-NPC	1 wt% Pd/Al_2_O_3_-120	180	7	6	100	100	5.43	[158]
12H-NPC	3 wt% Pd@MIL-101	190	0.1	4	100	100	5.43	[74]

^a^: Conversion of LOHCs (complete hydrogenation carriers); ^b^: yield of complete dehydrogenation products.

**Table 5 materials-16-03735-t005:** The performance of typical bimetallic catalysts for the dehydrogenation of LOHCs.

LOHCs	Catalysts	T (°C)	P (MPa)	Time (h)	Conv ^a^. (%)	Yield ^b^ (%)	H_2_ Release (wt%)	H_2_ Evolution Rate mmol/g_met_/min	Ref.
CYH	10 wt% Ni_0.8_Cu_0.2_/ACC	350	-	10	25.78	-	-	39.45	[167]
CYH	10 wt% Ag-1 wt% Pd/ACC	300	-	7	-	-	-	7.5	[168]
CYH	10 wt% Ag-1 wt% Rh/ACC	300	-	6	-	-	-	12.34	[168]
CYH	10 wt% Ag-1 wt% Pt/ACC	300	-	6	-	-	-	13.36	[168]
CYH	5 wt% 1:4 Ag-Rh/Y_2_O_3_	300	0.1	4	-	-	-	400	[80]
CYH	5 wt%1:4 Ag-Rh/ACC	300	0.1	5	-	-	-	178.7	[80]
MCH	2.5wt%Pt_0.8_Ir_0.2_/Mg-Al-O	350	-	1.6	91.1	99.9	-	263.9	[169]
MCH	2.0 wt% Pt-0.5 wt% Sn/MgAleO-350	300	-	12	90.5	-	-	262.1	[170]
MCH	Pt-Cu/S-1	400	0.1	6	92.26	-	-	445.3	[171]
12H-NEC	5 wt% PdCo/NGC	180	0.1	6	100	97.87	5.71	-	[172]
12H-NEC	Pd_3_ (3.75 wt%)-Ni_1_/SiO_2_	180	0.1	8	100	91.1	5.63	-	[173]
12H-NEC	Pd_3_ (3.75wt%)-Cu_1_/SiO_2_	180	0.1	8	100	83.11	5.47	-	[173]
12H-NEC	Pd_3_ (3.75wt%)-Au_1_/SiO_2_	180	0.1	8	100	94.9	5.7	-	[174]
12H-NEC	0.65 mol%Pd_1.3_–0.52 mol% Au1/rGO	180	0.1	4	100	100	5.79	-	[175]
12H-NEC	0.58 mol%Pd_1.3_–0.42 mol% Ru_1_/rGO	180	0.1	4	100	84.11	5.48	-	[175]
12H-NEC	Pd_1_ (2.5 wt%)-Co_1_/Al_2_O_3_	180	0.1	8	100	85.4	5.52	-	[176]
12H-NEC	Pd_4_Ni_1_/KIT-6	180	0.1	6	-	-	5.74	-	[177]
12H-NEC	Pd_1.2_Cu/rGO	180	0.1	7	100	100	5.79	-	[178]
12H-NPC	5 wt%Pd_1_-Ni_1_/Al_2_O_3_	180	6	7	100	100	5.43	-	[179]

^a^: Conversion of LOHCs (complete hydrogenation carriers); ^b^: yield of complete dehydrogenation products.

**Table 6 materials-16-03735-t006:** The performance of typical bifunctional catalysts of LOHCs.

LOHCs	Catalysts	Hydrogenation Reaction			Dehydrogenation Reaction			Ref.
		Reaction Conditions	Time (h)	H_2_ Uptake (wt%)	Reaction Conditions	Time (h)	H_2_ Release (wt%)	
NEC	Co-B/Al_2_O_3_-YH_3_	180 °C, 10 MPa H_2_	2	5.60	200 °C, 0.1 MPa H_2_	7	5.5	[68]
NEC	Pd/Al_2_O_3_-YH_3_	180 °C, 10 MPa H_2_	2	5.5	200 °C, 0.1 MPa H_2_	4	5.5	[189]
NPC	1 wt% Pd/CeO_2_-Al_2_O_3_	150 °C, 7 MPa H_2_	3	5.43	190 °C, 0.1 MPa H_2_	3	5.43	[192]
NEC	10 wt% LaNi_5.5_	180 °C, 7 MPa H_2_	4.5	5.5	200 °C, 0.1 MPa H_2_	4	5.5	[193]
NEC	0.52 mol% Pd_2_Ru@SiCN	110 °C, 2 MPa H_2_	36	5.68	180 °C, 0.1 MPa H_2_	7	5.51	[194]
NEC	0.6% Rh–1% Rd/γ-Al_2_O_3_	160 °C, 6 MPa H_2_	1	5.46	180 °C, 0.1 MPa H_2_	4	5.48	[191]
NPC	5 wt% Ru_0.5_Pd_0.5_/Al_2_O_3_	150 °C, 7 MPa H_2_	7	5.41	180 °C, 0.1 MPa H_2_	4	5.38	[67]
NEC	1 wt% Pd/LaNi_5_	180 °C, 7 MPa H_2_	0.7	5.5	200 °C, 0.1 MPa H_2_	2.1	5.5	[190]

## Data Availability

No new data were created.

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
