# Peer review of "Heterogeneous Catalysts in N-Heterocycles and Aromatics as Liquid Organic Hydrogen Carriers (LOHCs): History, Present Status and Future"

_materials, 2023, doi:10.3390/ma16103735_

Round 1
Reviewer 1 Report
The submitted manuscript by Jinxu Zhang, Fusheng Yang, Bin Wang, Dong Li, Min Wei, Tao Fang, and Zaoxiao Zhang describes an interesting state of the art of application of liquid organic hydrogen carriers (LOHC). The revision is attractive due to the potential of using hydrogen as fuel in a cycling way. Their study focused on the advances in the catalysts and the main variables that can improve the performance. The results are interesting, but I think that some details in the manuscript need to be improved before its publication.
Below are my recommendations:
In the introduction, the authors present an interesting discussion about the common compounds that have been used as hydrogen carriers. I think that would be recommended to describe the source of hydrogen used in those syntheses (hydrogenation). If the hydrogen used was obtained from fossil sources (e.g., coal, natural gas, or oils) or from conventional routes such as methane reforming it could be contradictory to the approach of this review.
Water electrolysis is proposed as a viable process to supply sufficient hydrogen to this system. However, I think the manuscript should include a brief analysis to describe whether that process is a techno-economic viable alternative.
I recommend adding an analysis of reactions thermodynamics (hydrogenation and dehydrogenation) because considering the thermodynamic constraints is crucial to understand the viability of the proposed catalytic systems.
The manuscript shows interesting results with the use of rare earth as promoters in the catalysts. I recommend including a critical discussion about the use of these elements (due to their availability) for a possible industrial application.
Line 247. The authors describe the activity of a Ru catalyst in both reactions, hydrogenation, and dehydrogenation. Since this is the first example of using a single catalyst in both reactions, I think more details about the activity and selectivity of the dehydrogenation step must be added.
Section 2.1.1. The authors describe the use of different types of catalysts in the hydrogenation reaction. I recommend including at the final of this section an analysis (from their critical review) of the variables (properties) that have significant contributions. This could help readers to understand the specific advances and how could improve the design of the catalysts. Also, I recommend adding a discussion of the catalysts’ stability.
Between lines 239-242, the authors discuss a new reported reaction pathway. However, the details of the used characterization technique are unclear. In addition, I would recommend changing the reference Figure 4 (line 272) because physical evidence of that mechanism is not included.
Lines 323-324. To add details of how the electron deficiency in the Pd sites was evaluated in the cited report.
Lines 446-453. What is the physical evidence that supports the reaction mechanism shown in the cited reference?
Line 481. There is a mistake in the subsection title, I think it should be “dehydrogenation.”
Lines 590-592. To add details of the technique used to assess the metal-support interaction.
Line 628. The title and numeration of the subsection are wrong. I think they should be read as “2.2.2” and “dehydrogenation.”
Lines 656-658. To include the characterization results that demonstrate the electron transfer between Sn and Pt. I would recommend changing the Figure of reference (Figure 11) because evidence of the electron transfer is not included.
Section 3. I recommend adding a general diagram (as an example) of the reaction system. This could help readers to understand how the samples are obtained from the system to further analysis, how the catalytic activity is evaluated, and how the reaction conditions are changed to finish the hydrogenation and start the dehydrogenation.
Lines 805-808. Indicate how the oxygen vacancies were identified.
Author Response
Dear reviewer, we are pleased to receive your response and have made corresponding modifications based on your comments and suggestions.
Comments and Suggestions 1: “I think that would be recommended to describe the source of hydrogen used in those syntheses (hydrogenation). If the hydrogen used was obtained from fossil sources (e.g., coal, natural gas, or oils) or from conventional routes such as methane reforming it could be contradictory to the approach of this review.”
Answer 1: Thank you for your valuable suggestions. The catalysts mentioned in our review are mostly in the laboratory stage. In the catalytic reactions of these catalysts, the hydrogen used in the hydrogenation reaction of LOHCs is mostly purchased from reagent companies. We do not know the specific sources of these hydrogen and the references have not been reported. Therefore, we apologize for not being able to provide information in this regard. Liquid organic hydrogen carrier technology can be used to store hydrogen generated from renewable energy sources, which has broader application prospects in the future.
Comments and Suggestions 2: “However, I think the manuscript should include a brief analysis to describe whether that process is a techno-economic viable alternative.”
Answer 2: Thank you for your valuable suggestions. We have provided a more detailed description of the production methods of hydrogen (lines 33-46). Considering energy sources and energy conversion efficiency, we believe that green hydrogen generated from renewable energy is a promising hydrogen preparation method that can become a feasible hydrogen supply solution in the future.
Comments and Suggestions 3: “I recommend adding an analysis of reactions thermodynamics (hydrogenation and dehydrogenation) because considering the thermodynamic constraints is crucial to understand the viability of the proposed catalytic systems..”
Answer 3: Thank you for your valuable suggestions. The thermodynamic description of hydrogenation and dehydrogenation reactions has been supplemented (lines 107-117).
Comments and Suggestions 4: “The manuscript shows interesting results with the use of rare earth as promoters in the catalysts. I recommend including a critical discussion about the use of these elements (due to their availability) for a possible industrial application.”
Answer 4: Thank you for your valuable suggestions. The corresponding content has been added to the end of Section 2.1.1(lines 318-325)
Comments and Suggestions 5: “Since this is the first example of using a single catalyst in both reactions, I think more details about the activity and selectivity of the dehydrogenation step must be added.”
Answer 5: Thank you for your valuable suggestions. We apologize for this misread. What we mean is that the dehydrogenation reaction of cis products is prone to occur on the surface of the dehydrogenation catalyst, rather than on the Ru/YH3 surface. The Ru/YH3 catalyst only has catalytic hydrogenation function. Our expression is ambiguous and we have corrected it.
Comments and Suggestions 6: “Section 2.1.1. The authors describe the use of different types of catalysts in the hydrogenation reaction. I recommend including at the final of this section an analysis (from their critical review) of the variables (properties) that have significant contributions. This could help readers to understand the specific advances and how could improve the design of the catalysts. Also, I recommend adding a discussion of the catalysts’ stability.”
Answer 6: Thank you for your valuable suggestions. The corresponding content has been added to the end of Section 2.1.1(lines 326-332). We also mentioned the content of catalyst stability in the review and we have made appropriate supplements. (lines 167,245,309)
Comments and Suggestions 7: “Between lines 239-242, the authors discuss a new reported reaction pathway. However, the details of the used characterization technique are unclear. In addition, I would recommend changing the reference Figure 4 (line 272) because physical evidence of that mechanism is not included.”
Answer 7: Thank you for your valuable suggestions. We have added the validation experiment content of the original author (lines 265-270). If you want to know more detailed information, you can refer to the cited literature. Besides, We have replaced and added characterization evidence(Figure 4, lines 336).
Comments and Suggestions 8: “Lines 323-324. To add details of how the electron deficiency in the Pd sites was evaluated in the cited report.”
Answer 8: Thank you for your valuable suggestions. The corresponding content has been added to the end of Section 2.1.2(lines 359-363). If you want to know more detailed information, you can refer to the cited literature.
Comments and Suggestions 9: “Lines 446-453. What is the physical evidence that supports the reaction mechanism shown in the cited reference?”
Answer 9: Thank you for your valuable suggestions. Some physical evidences that supports the reaction mechanism was added (lines 491-496). Due to the extensive descriptions in the original literature, we have incorporated some of the main characterization results into our review. For more detailed information, please refer to the original literature.
Comments and Suggestions 10: “Line 481. There is a mistake in the subsection title, I think it should be “dehydrogenation.”
Answer 10: Thank you for your valuable suggestions. We have corrected this mistake.
Comments and Suggestions 11: “To add details of the technique used to assess the metal-support interaction.”
Answer 11: Thank you for your valuable suggestions. We have added some information on techniques used to evaluate metal support interactions(lines 640-643). Due to the numerous characterization results in the original literature, please refer to the original literature for more detailed information.
Comments and Suggestions 12: “The title and numeration of the subsection are wrong. I think they should be read as “2.2.2” and “dehydrogenation.”
Answer 12: Thank you for your valuable suggestions. We have corrected this mistake.
Comments and Suggestions 13: “Lines 656-658. To include the characterization results that demonstrate the electron transfer between Sn and Pt. I would recommend changing the Figure of reference (Figure 11) because evidence of the electron transfer is not included.”
Answer 13: Thank you for your valuable suggestions. Figure 11 is a schematic diagram of the catalyst and we have modified its reference position (lines 696). The XP spectra of Sn 3d demonstrated the electron transfer mechanism between Sn and Pt. The original article contains much related characterization and analysis content, so we only retained its conclusion. For more detailed information, please refer to the original literature.
Comments and Suggestions 14: “Section 3. I recommend adding a general diagram (as an example) of the reaction system. This could help readers to understand how the samples are obtained from the system to further analysis, how the catalytic activity is evaluated, and how the reaction conditions are changed to finish the hydrogenation and start the dehydrogenation.”
Answer 14: Thank you for your valuable suggestions. The schematic diagram(Figure 14) and corresponding description (lines 827-842) of the catalytic reaction evaluation device have been added to Section 3.
Comments and Suggestions 15: “Lines 805-808. Indicate how the oxygen vacancies were identified.”
Answer 15: Thank you for your valuable suggestions. The corresponding content has been added to Section 3 (lines 876-881). If you want to know more detailed information, you can refer to the cited literature.
Reviewer 2 Report
The paper was revised according to the journal rules.
Few revisions are required and they are reported below:
- please add a nomenclature list for all acronyms and parameters
- I suggest to improve the quality of figures reported
- hydrogen as energy carrier must be produced by renewable energies, clarify this aspect also in the abstract
- hydrogen production should be detailed and discussed considering the primary source in terms of energy required and efficiency of conversion
- i suggest to add the energy requirements for the gas compression storage system
- Repeatability of measurements reported in figure 1 should be added
- data of ciclability and performance should be added also for figure 2
- the efficiency decay of the hydrogen storage should be considered
- maintenance aspects could be considered
- coke deposits and performance decay should be detailed
- future aspects could be more detailed in the conclusion section
Author Response
Dear reviewer, we are pleased to receive your response and have made corresponding modifications based on your comments and suggestions.
Comments and Suggestions 1: “please add a nomenclature list for all acronyms and parameters”
Answer 1: Thank you for your valuable suggestions, the nomenclature list for all acronyms and parameters is added to the end of the article (page 30 and 31).
Comments and Suggestions 2: “ I suggest to improve the quality of figures reported”
Answer 2: Thank you for your valuable suggestions. We have replaced the original image with a high-resolution image.
Comments and Suggestions 3: “hydrogen as energy carrier must be produced by renewable energies, clarify this aspect also in the abstract”
Answer 3: Thank you for your valuable suggestions. The corresponding description has been added to the summary section as required, with the specific changes as follows “Hydrogen generated from renewable energy sources is considered as a promising energy carrier”.
Comments and Suggestions 4: “hydrogen production should be detailed and discussed considering the primary source in terms of energy required and efficiency of conversion”
Answer 4: Thank you for your valuable suggestions. The corresponding content on hydrogen production has been added to the introduction as required (line 33-46).
Comments and Suggestions 5: “i suggest to add the energy requirements for the gas compression storage system”
Answer 5: Thank you for your valuable suggestions. The energy requirements for the gas compression storage system is added to the introduction.(line 62)
Comments and Suggestions 6: “Repeatability of measurements reported in figure 1 should be added”
Answer 6: Thank you for your valuable suggestions. Figure 1 is a schematic diagram from the original paper. We have not found repeatability of measurements dates in the original article, so we regret not being able to provide additional content in this regard.
Comments and Suggestions 7: “data of culpability and performance should be added also for figure 2”
Answer 7: Thank you for your valuable suggestions. Data of culpability and performance has been added to figure 2.
Comments and Suggestions 8: “the efficiency decay of the hydrogen storage should be considered”
Answer 8: Thank you for your valuable suggestions. During multiple cycles of hydrogen storage, the hydrogen storage efficiency will decrease as the performance of the catalyst decreases, which is reflected in the cyclic stability of the catalyst. This part is mentioned in our article. For details, please refer to lines 167, 245, 309,397.ect.
Comments and Suggestions 9: “maintenance aspects could be considered”
Answer 9: Thank you for your valuable suggestions. Currently, most of the catalysts involved in our article are in the laboratory stage, and the corresponding original articles lack maintenance reports, so we can’t be able to supplement this aspect.
Comments and Suggestions 10:coke deposits and performance decay should be detailed
Answer 10: Thank you for your valuable suggestions, The corresponding content has been added to page 19, lines 684-686
Comments and Suggestions 11: “future aspects could be more detailed in the conclusion section”
Answer 11: Thank you for your valuable suggestions. The future aspects has been appropriately supplemented and modified.
Reviewer 3 Report
The review focuses on the catalysts used for the hydrogenation and dehydrogenation of LOHC couples. As it is exclusively dedicated to N-heterocycles and Aromatics (other LOHC are not cited), and only concerns heterogeneous catalysts, the title has to be changed accordingly.
The paper is divided into 3 sections : 1) catalysts for hydrogenation, 2) catalysts for dehydrogenation and 3) catalysts adapted to both reactions.
In table 3, write all the Times in h (and not some in min). In table 4, write all the pressures in MPa (and not some in kPa). Write 0.1 instead of n.p. and instead of 101kPa.
The paragraph lines 895-899 is wrong. Only the hydrogenations are exothermic. Please check and modify.
Considering these modifications, the paper deserves publication in Materials.
Typos:
- l. 52 (re-action)
- l. 82 (N.)
- l. 86 (is-sue)
- l. 90 (reaction)
- l. 94 (con-trolled)
- l. 152 (with the catalytic activity remains)
- l. 170 (of in)
- l. 209 (The authors that)
- l. 590 (T The)
- l. 649 comma instead of point
- l. 682 (Anaam H et al.)
Author Response
Dear reviewer, we are pleased to receive your response and have made corresponding modifications based on your comments and suggestions.
Comments and Suggestions 1: “the title has to be changed accordingly”
Answer 1: Thank you for your valuable suggestions. We have changed the title to “Heterogeneous Catalysts in N-Heterocycles and Aromatics as Liquid Organic Hydrogen Carriers (LOHCs): History, Current Status, and Future”
Comments and Suggestions 2: “The paper is divided into 3 sections : 1) catalysts for hydrogenation, 2) catalysts for dehydrogenation and 3) catalysts adapted to both reactions.”
Answer 2: We sincerely thank you for your valuable suggestions. We divide the catalysts into single-function and bifunctional parts to be able to strictly distinguish different catalysts. We believe that the catalysts for hydrogenation will overlap with catalysts adapted to both reactions to a certain extent. Catalysts adapted to can also be understood as a part of catalysts for hydrogenation. So we think it may be better to divide the catalysts into single-function and bifunctional parts.
Comments and Suggestions 3: “In table 2, write all the Times in h (and not some in min). In table 4, write all the pressures in MPa (and not some in kPa). Write 0.1 instead of n.p. and instead of 101kPa. ”
Answer 3: Thank you for your valuable suggestions. We have corrected and harmonized the data units in Tables 3 and 4.
Comments and Suggestions 4: “The paragraph lines 895-899 is wrong. Only the hydrogenations are exothermic. Please check and modify.”
Answer 4: Thank you for your valuable suggestions. The description in lines 895-899 is based on Figure 16 (c). From Figure 16 (c), it can be seen that the hydrogenation process from 4H-NEC to 6H-NEC and from 8H-NEC to 12H-NEC is endothermic. According to the reversibility of the hydrogenation and dehydrogenation processes, we describe the dehydrogenation process from 12H-NEC to 8H-NEC and from 6H-NEC to 4H-NEC as exothermic. However, due to the controversy over whether the dehydrogenation processes from 12H-NEC to 8H-NEC and from 6H-NEC to 4H-NEC are exothermic, we have decided not to use this image anymore and modify the corresponding description.
Comments and Suggestions 5: “Typos:.”
Answer 5: Thank you for pointing out the typos and we have corrected them all.
Round 2
Reviewer 1 Report
The authors answered all the concerns and comments positively. I think that this version of the manuscript is improved; therefore, I recommend publishing it in this form
Reviewer 2 Report
I suggest improving the quality of the figures, they are somewhat blurred
Reviewer 3 Report
The authors have taken into account all the remarks done. The article is suitable for publication in Materials.
Please, for the final version, take into account the following typos:
- in Table 3, convert 700 psi in MPa
- l. 967 are easily occur